# Substantial trace metal input from the 2022 Hunga Tonga-Hunga Ha'apai eruption into the South Pacific

Zhouling Zhang [1] ✉, Antao Xu [1,4], Ed Hathorne[1], Marcus Gutjahr [1], Thomas J. Browning [1], Kathleen J. Gosnell[1], Te Liu[1,5], Zvi Steiner [1], Rainer Kiko[1,2], Zhongwei Yuan [1,3], Haoran Liu[1,3], Eric P. Achterberg [1] & Martin Frank [1]

The January 2022 eruption of the Hunga Tonga-Hunga Ha'apai (HTHH) volcano discharged 2,900 teragrams of ejecta, most of which was deposited in the South Pacific Ocean. Here we investigate its impact on the biogeochemistry of the South Pacific Gyre (SPG) using samples collected during the GEOTRACES cruise GP21 in February-April 2022. Surface water neodymium isotopes and rare earth element compositions showed a marked volcanic impact in the western SPG, potentially extending to the eastern region. Increasing trace metal concentrations in surface waters and chlorophyll-a inventories in euphotic layers between the eastern and western SPG further suggest that the volcanic eruption supplied (micro)nutrients potentially stimulating a biological response. We estimate that the HTHH eruption released up to 0.16 kt of neodymium and 32 kt of iron into the SPG, which is comparable to the annual global dust-borne Nd flux and the annual dust-borne Fe flux to the entire SPG, respectively.

Volcanic ash deposited into the surface ocean readily releases bioactive trace metals (e.g., iron, Fe; manganese, Mn), which can stimulate phytoplankton growth, either directly via Fe fertilization of waters replete in macronutrients, or indirectly via Fe fertilization of nitrogen fixation in waters where nitrate is depleted and phosphate available in excess[1–3]. Consequently, major volcanic eruptions driving surface ocean Fe fertilization have been linked to atmospheric $CO_2$ drawdown and climatic changes in the past[4–7]. Airborne volcanic ash has been suggested as an important atmospheric Fe source for the surface Pacific Ocean, due to the abundance of active and explosive volcanoes surrounding it (Pacific Ring of Fire)[3]. The estimated input of ash into the Pacific Ocean of $128–221 \times 10^{12}$ g/yr, is comparable to estimates of mineral dust inputs of non-volcanic origin ($39–519 \times 10^{12}$ g/yr)[8], resulting in a total soluble Fe flux of $3–75 \times 10^6$ mol/yr to the Pacific Ocean[3]. Indeed, dispersed ash forms a substantial fraction of the South Pacific Ocean (SPO) pelagic clay deposits, often exceeding 50% by

mass, and records episodes of southern hemisphere volcanism[3,9]. The SPO is dominated by the vast oligotrophic South Pacific Gyre (SPG), which is characterized by extremely low concentrations of nitrate, relatively elevated phosphate, and expected Fe limitation of nitrogen fixation rates[10].

The underwater Hunga Tonga-Hunga Ha'apai (HTHH) volcano (20.536°S, 175.382°W; 150 m below sea-level), located in the Tonga-Kermadec volcanic arc, erupted violently on 15 January 2022 (Fig. 1a). The eruption is among the largest volcanic activity ever recorded with modern geophysical instrumentation, with a volcanic explosivity index (VEI) of ~6[11]. The eruption produced a large volcanic plume that reached an altitude of up to 57 kilometers[12] and resulted in tephra covering an area of at least 600 km in diameter[13]. The eruption expelled vast ($2.9 \times 10^{15}$ g) amounts of basaltic-andesitic volcanogenic material, such as ash and pumice, into the SPG[14]. Following the deposition of the ash-laden volcanic plumes of HTHH, satellite

[1]GEOMAR Helmholtz Centre for Ocean Research Kiel, Kiel, Germany. [2]Faculty of Mathematics and Natural Sciences, Kiel University, Kiel, Germany. [3]State Key Laboratory of Marine Environmental Science and College of Ocean and Earth Sciences, Xiamen University, Xiamen, China. [4]Present address: Institute of Environmental Physics, Heidelberg University, Heidelberg, Germany. [5]Present address: School of Ocean and Earth Science, University of Southampton, Southampton, UK. ✉e-mail: zzhang@geomar.de

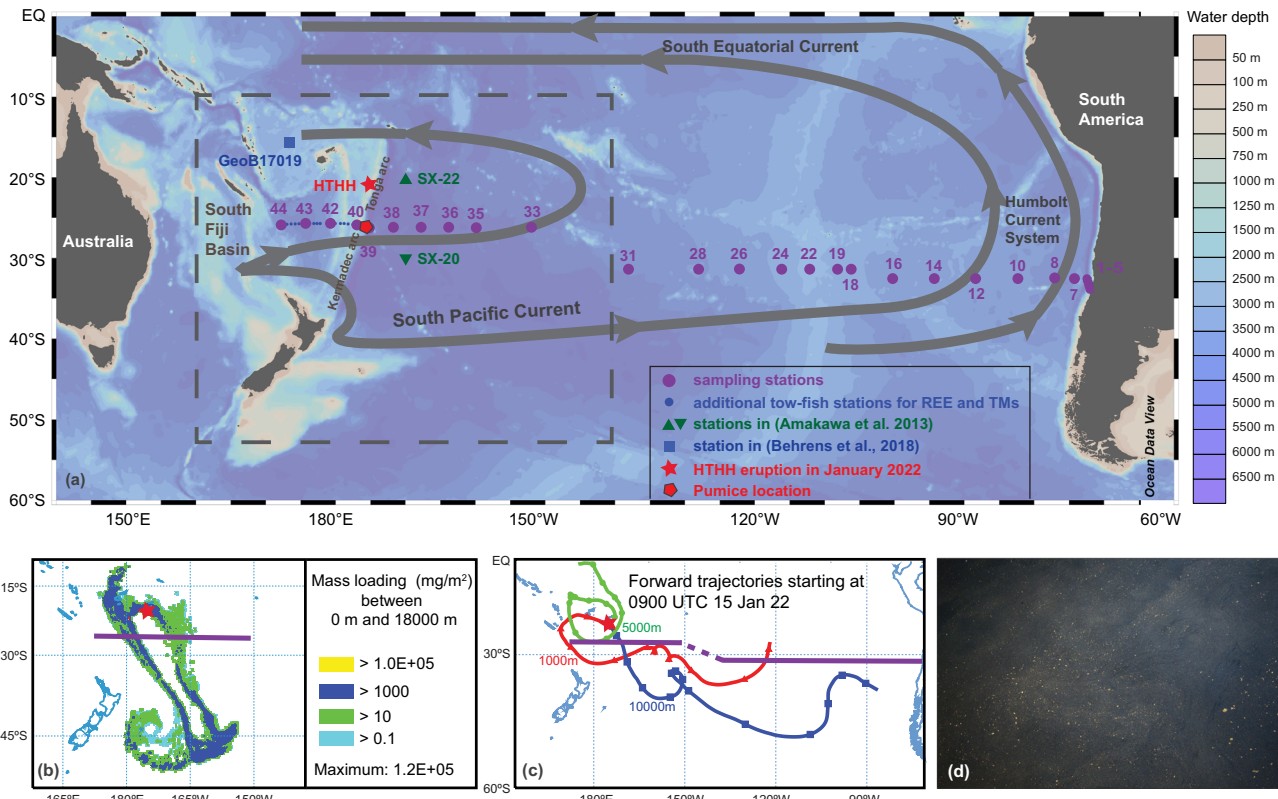

**Fig. 1 | GEOTRACES GP21 cruise track and volcanic ash dispersion from the Hunga Tonga-Hunga Ha'apai (HTHH) volcanic eruption. a** Map showing the sampling stations from this and previous studies[36,37], the locations of the HTHH volcanic eruption, and the pumice encountered during the cruise. Additional tow-fish samples for rare earth elements (REE) and trace metal (TM) measurements were taken between stations 40 and 44. Surface currents in the South Pacific Ocean are adapted from Figure 10.1 in ref. 32. The map is generated using Ocean Data View[80]. **b** Volcanic ash particle deposition (mg/m²) between 0 and 18000 meters after 72 h of the HTHH volcanic eruption based on the NOAA HYSPLIT volcanic ash dispersion model[67], shown in the area enclosed by the dashed square box in subplot a. Mass loadings after 24 and 48 h of the eruption are shown in Supplementary

Fig. 3. **(c)** Air mass forward trajectories at 1000 (red), 5000 (blue), and 10000 (green) meters above model ground level for 315 hours after the HTHH eruption, based on the NOAA HYSPLIT trajectory model[67]. Forward trajectories at 20, 100, and 500 meters post-eruption are shown in Supplementary Fig. 4. Model details and parameterization of both b and c are given in the Methods section. Our cruise track is depicted by a purple line in subplots b and c. **(d)** Photograph of floating pumice, with sizes up to several centimeters, taken from the research vessel during the cruise before crossing the Tonga-Kermadec Ridge on 30 March 2022. Photo credit: Maria de los Angeles Amenabar. Additional photos of floating pumice and individual pumice that were collected can be found in Supplementary Fig. 2.

observations revealed elevated surface water chlorophyll-a concentrations (Chl-a) in the affected nearby region[15,16].

Volcanic ash from the HTHH eruption has the potential to exert widespread effects on the SPG via long-distance (thousands of kilometers) atmospheric transport by prevailing winds[17–19]. Although suspended ash was still observed in the water column around the HTHH volcano months later[20], the volcanic ash overall sank rapidly (hours to days) through the water column, depending on particle size[21]. In contrast, pumice can float for periods of months to years due to its high porosity and trapped gas bubbles and can thus be transported over thousands of kilometers via surface ocean currents[22,23]. The dispersal of volcanic ash and floating pumice can enhance the impacts of a volcanic eruption on surface ocean biogeochemistry over extended temporal and spatial scales[23]. However, so far, the large-scale impact of the HTHH eruption on SPG biogeochemistry has not been explored.

Tracing volcanic inputs into the SPG is challenging due to the difficulty in distinguishing them from other external sources such as eolian aluminosilicate dust derived from continents[24] and weathering inputs from volcanic islands[25]. However, radiogenic neodymium isotope compositions (¹⁴³Nd/¹⁴⁴Nd, expressed as $\varepsilon_{Nd}$) and dissolved rare earth element (REE) concentrations in seawater can be used to trace specific sources of trace elements to the ocean[26–28]. The $\varepsilon_{Nd}$ signatures of lithogenic materials vary as a function of age and lithology and surface waters acquire their $\varepsilon_{Nd}$ signatures via aeolian input, volcanic

input, and continental weathering input from rivers and submarine groundwater discharge[26,29]. The REEs are divided into three categories based on their atomic weight: light (LREE), medium (MREE), and heavy (HREE). To eliminate natural abundance variations, REE concentrations in seawater are normalized to Post-Archean Australian Shale (PAAS) which approximates average continental crust[30]. The REE patterns in the open ocean increase in abundance from LREE to HREE because of preferential LREE scavenging in seawater, which is known as HREE enrichment (HREE/LREE$_{PAAS}$ > 1)[31]. The REEs also typically exhibit a cerium (Ce) depletion in comparison to neighboring REEs with similar mass due to Ce oxidation to insoluble forms in seawater, which is referred to as the "Ce anomaly" (Ce/Ce*$_{PAAS}$ < 1)[31]. The dissolution of lithogenic material weakens the Ce anomaly (Ce/Ce*$_{PAAS}$ closer to 1) and modifies the dissolved HREE/LREE ratio, depending on the composition of the dissolving material.

To investigate the biogeochemical impacts of the HTHH eruption on the surface SPG, we characterized the magnitude and location of the ash deposition using a volcanic ash dispersal model and an air mass trajectory model. The volcanic inputs were traced with dissolved $\varepsilon_{Nd}$ and REE signals in surface seawater and biogeochemical impacts were examined via the concentrations of trace metals (dissolved aluminum (Al), Mn, Fe) and chlorophyll-a, which is a proxy for phytoplankton biomass. Samples were collected along a zonal transect at 26-32.5°S across the entire SPO during GEOTRACES cruise GP21 in February-April

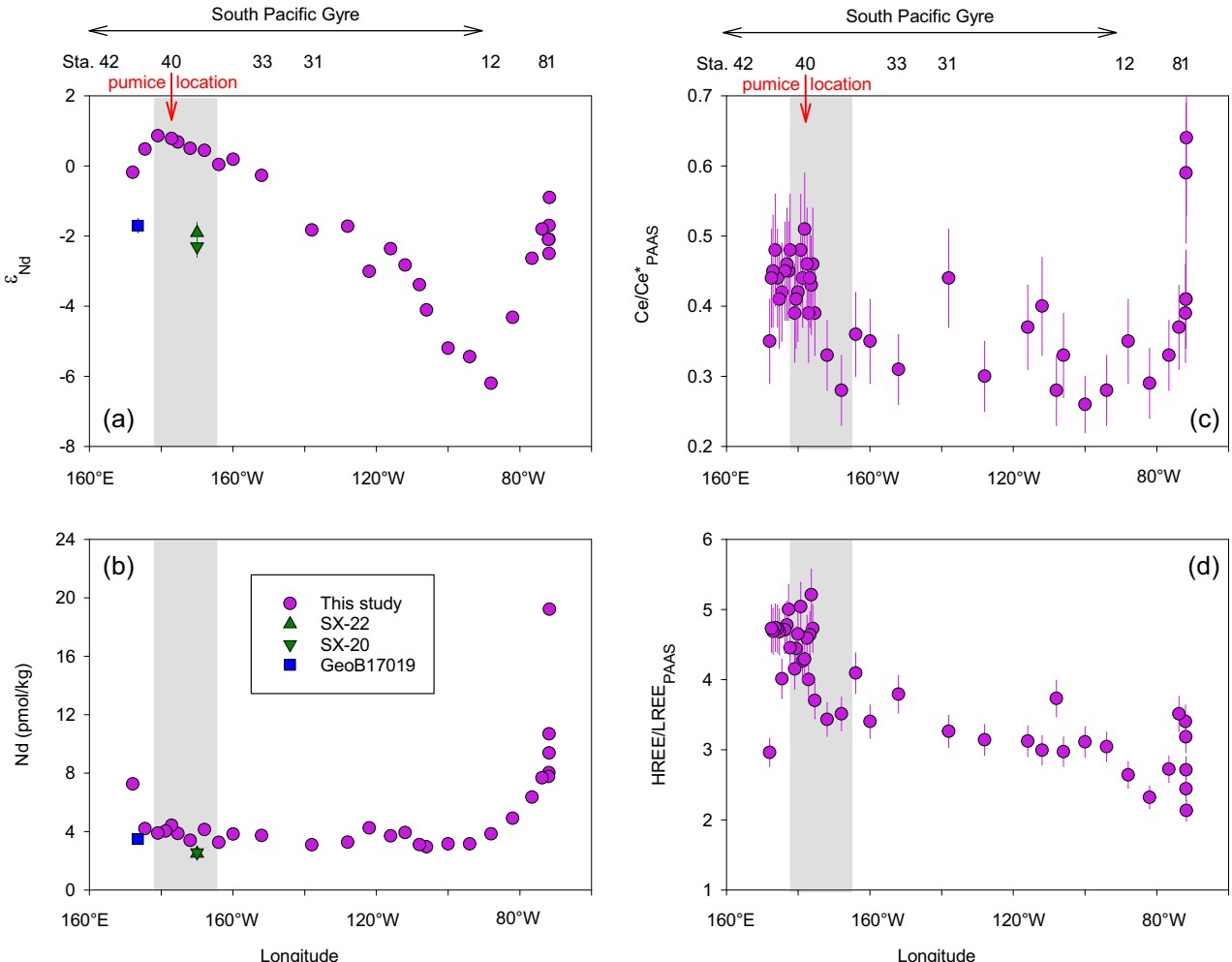

**Fig. 2 | Changes in Nd isotopic composition ($\varepsilon_{Nd}$) and concentration, and selected Rare Earth Elements (REEs) ratios in the surface water (uppermost 3-5 m) along GP21. a** $\varepsilon_{Nd}$, **b** Nd concentration measured by isotope dilution method, **c** Post-Archean Australian Shale (PAAS) normalized Ce anomaly (Ce/Ce*$_{PAAS}$), calculated as $2 \times Ce_{PAAS}/(La_{PAAS}+Pr_{PAAS})$, **d** PAAS normalized heavy REE enrichment relative to light REE (HREE/LREE$_{PAAS}$), calculated as $(Yb_{PAAS}+Lu_{PAAS})/(Pr_{PAAS}+Nd_{PAAS})$. Data from this study are shown in pink. The location of the pumice is indicated by a red arrow pointing downwards. The region of ash deposition within 3 days of the eruption, based on the NOAA HYSPLIT volcanic ash dispersion model (Fig. 1b, Supplementary Fig. 3), is shown in the transparent gray box. In a and b, the external 2 SD of the $\varepsilon_{Nd}$ values and Nd concentrations are smaller than symbol size. Previously reported data for station SX-22 (20°S, 170°W)[36] are shown in green triangle up, station SX-20 (30°S, 170°W)[36] in green triangle down, and station GeoB17019 (15°S, 174°E)[37] in blue square. In c and d, the error bars are derived from the propagation of measurement errors (2 SD) of each element.

2022 (Fig. 1a). We find that the western SPG exhibited strong (bio)geochemical impacts following the volcanic inputs originating from the HTHH, which likely extended to the central SPG via atmospheric ash dispersal and surface current transport.

## Results and discussion

### Expedition GP21 and the HTHH eruption

The Pacific GEOTRACES GP21 expedition (Chile to New Caledonia) with the research vessel SONNE (SO289) took place between 23 February and 4 April 2022, which was 39 to 79 days after the HTHH eruption (Fig. 1a). The transect was aligned with the southern limb of the anticyclonic SPG, which is formed by the broad, eastward-flowing South Pacific Current[32]. Underway data from the upper ocean Acoustic Doppler Current Profiler (75 kHz-ADCP) showed that the prevailing surface current velocities were eastward, with typical surface speeds of 10-40 cm per second (Supplementary Fig. 1). However, a small region (166-174°W) in the vicinity of the Tonga-Kermadec Ridge exhibited predominantly westward surface currents. A large amount of floating tephra, mostly pumice, was observed and collected before crossing the Tonga-Kermadec Ridge

(-175°W) on 30 March 2022, 73 days after the major HTHH eruption (Fig. 1a, d; Supplementary Fig. 2).

The NOAA HYSPLIT volcanic ash dispersion model indicated that volcanic ash particles were mainly deposited in the western region of the SPG (20°-50°S, 175°E -150°W), south of the HTHH volcano, within 3 days of the eruption (Fig. 1b; Supplementary Fig. 3). This area covered the western extent of the GP21 transect. NOAA HYSPLIT forward air mass trajectories for 13 days following the HTHH eruption indicated a general eastward atmospheric transport of volcanic ash across the entire SPG (Fig. 1c; Supplementary Fig. 4), which overlapped with the GP21 cruise track.

### Radiogenic Nd isotopic compositions and REE patterns in surface waters

Surface seawater $\varepsilon_{Nd}$ values exhibited pronounced and systematic variability (Fig. 2a). The $\varepsilon_{Nd}$ values decreased from −0.9 at the Chilean coast to −6.2 at the eastern boundary of the SPG (90°W), followed by a gradual increase to +0.9 near the Tonga-Kermadec Arc. The most unradiogenic (negative) $\varepsilon_{Nd}$ signal (−6.2) at 90°W is consistent with a reported surface value at a nearby location ($\varepsilon_{Nd}$ = −6.4; station EGY:

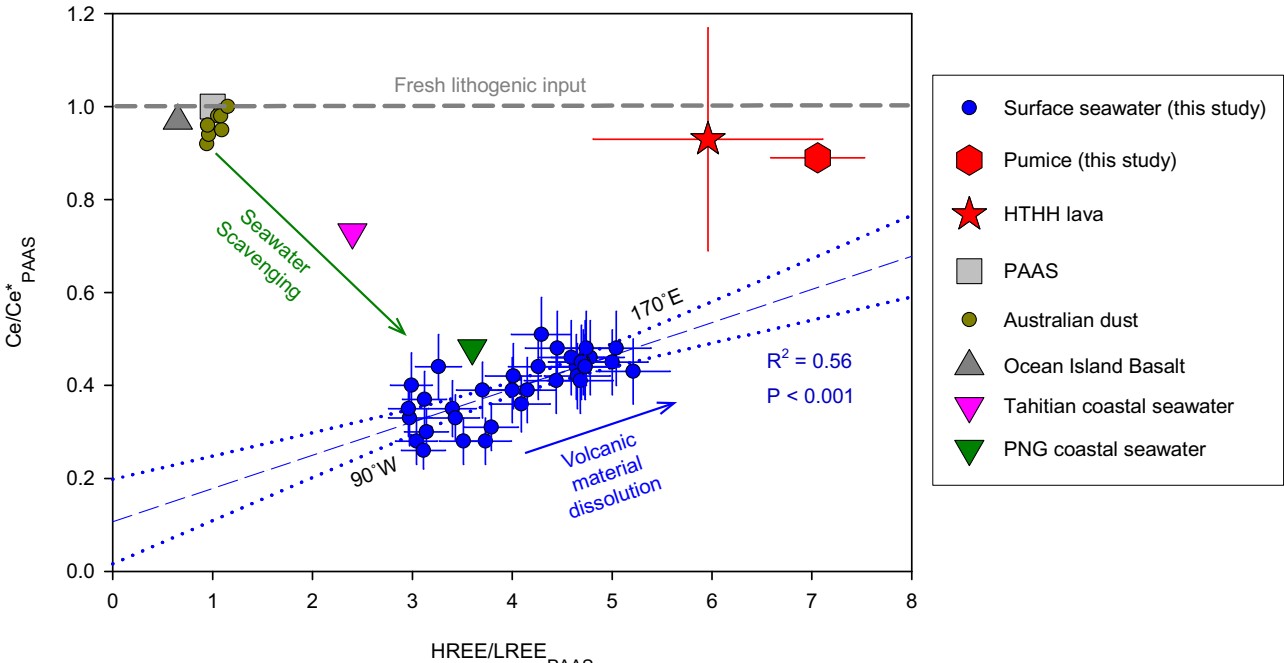

**Fig. 3 | Cross plot of Post-Archean Australian Shale (PAAS) normalized Ce anomaly (Ce/Ce*$_{PAAS}$) and heavy Rare Earth Element (REE) enrichment relative to light REE (HREE/LREE$_{PAAS}$) in the surface water (uppermost 3-5 m) of the South Pacific Gyre (170°E-90°W).** The REE data of the collected pumice, Hunga Tonga-Hunga Ha'apai (HTHH) lava[42], PAAS, Australian dust[81,82], Ocean Island Basalt (OIB)[40] and coastal seawater of two representative volcanic islands in the Pacific Ocean, Papua New Guinea (PNG)[28] and Tahiti[38], are included. The calculations for Ce/Ce*$_{PAAS}$ and HREE/LREE$_{PAAS}$ are consistent with those in Fig. 2. Australian dust data contains data from various major dust sources from Australia, namely the Murray–Darling Basin[81], Queensland, Paroo, Eyre Peninsula, and Channel Country[82]. The green arrow represents the change in REE patterns from OIB[40] to Tahitian coastal surface seawater (sample ID: t324)[38] and PNG coastal surface seawater near Sepik River mouth (station EUC-Fe-28)[28] due to seawater scavenging. The blue arrow indicates the change of REE patterns in surface seawater from eastern to western South Pacific Gyre as a result of volcanic material dissolution. The error bar for HTHH lava represents the standard deviation (SD) of five lava samples, while the error bars for surface water and pumice are derived from the propagation of measurement errors (2 SD) of each element. The dashed blue line and dotted blue lines represent the linear regression of Ce/Ce*$_{PAAS}$ and HREE/LREE$_{PAAS}$ in the surface water of the South Pacific Gyre (this study) and the 95% confidence interval of the regression, respectively.

32°S, 91.4°W)[33] and reflects the influence of surface currents. Specifically, as the South Pacific Current approaches South America, the Humboldt Current system between the SPG and the Chilean coast transports subantarctic water equatorward at approximately 90°W[34]. The surface water in the Subantarctic Zone of the southeastern Pacific Ocean carries an ε$_{Nd}$ signal ranging between −5.7 and −8.4[35] resulting in the unradiogenic ε$_{Nd}$ signal at ~90°W along the GP21 section. The surface ε$_{Nd}$ signal became more radiogenic (positive) between 90°W (−6.2) and the Chilean coast (−0.9; Fig. 2a). This was accompanied by a pronounced increase in Nd concentration from ~4 to ~20 pmol/kg (Fig. 2b) and a weakening of the Ce anomaly towards the Chilean coast (Fig. 2c). Collectively these geochemical seawater signatures were due to lithogenic input from the Chilean coast, where outcropping continental rocks and sediments exhibit highly radiogenic ε$_{Nd}$ signatures (0 to +5) due to their volcanic nature[33].

Within the SPG, the surface water ε$_{Nd}$ signal became more radiogenic from the eastern edge of the gyre at 90°W (−6.2) towards the Tonga-Kermadec Arc at 180°W (+0.9) (Fig. 2a). The ε$_{Nd}$ signature in the western SPG (~0 to +1) was more radiogenic than reported surface ε$_{Nd}$ signatures at nearby locations. This includes two stations at 170°W (SX-22: 20°S; SX-20: 30°S) sampled in January 2005 (ε$_{Nd}$ = −1.9 and −2.3)[36] and one station at 174°E (GeoB17019: 15°S) sampled in September-October 2012 (ε$_{Nd}$ = −1.7)[37]. The more positive values observed in our study indicate pronounced external inputs into the surface ocean prior to our sampling in the western SPG that were enriched in radiogenic ε$_{Nd}$. However, surface water Nd concentrations in the western SPG remained consistently low (~4 pmol/kg), at levels comparable to previous measurements[36,37] (Fig. 2b). It can be posited that scavenging and/or exchange processes occurred to maintain a constant Nd concentration while modifying the Nd isotopic composition. This will be discussed in greater detail in a later section (Rapid release and scavenging of Nd following the volcanic input). Accompanying the ε$_{Nd}$ increases, the Ce anomaly weakened (Fig. 2c) while the HREE enrichment strengthened (Fig. 2d) from eastern to western SPG.

**Potential sources of external inputs to SPG surface waters**
The positive correlation between Ce anomaly and HREE enrichment values ($R^2 = 0.56$, $p < 0.05$; Fig. 3) in the SPG (170°E-90°W) indicates that the radiogenic source of external inputs in the western SPG were characterized by a weaker (or absent) Ce anomaly and stronger HREE enrichment. Possible external source materials for the surface water of the western SPG include volcanic material of the HTHH eruption, such as ash or pumice, weathering of basaltic volcanic islands, aeolian dust from Australia, and/or shallow hydrothermal inputs along the Tonga-Kermadec Arc, each of which have particular ε$_{Nd}$ signatures and REE ratios (Table 1).

The dissolution of volcanogenic materials and also the weathering of basaltic volcanic islands[38] result in the release of Nd with a highly radiogenic signature. This is because oceanic crust, such as mid-ocean ridge basalts (MORBs), ocean island basalts (OIBs), and island arc rocks all have positive ε$_{Nd}$ values between ~0 and +10[39]. Consistent with that, our pumice sample collected near the Tonga-Kermadec arc yielded a highly radiogenic ε$_{Nd}$ signature of +7.5. The weathering of volcanic islands, for example Papua New Guinea, also supplies radiogenic Nd, resulting in ε$_{Nd}$ values of both surface and subsurface waters in the equatorial Western Pacific reaching up to +0.7[37]. However, basalts formed in different tectonic settings have distinct chemical characteristics, primarily determined by the source type, tectonic

**Table 1 | Nd isotopic composition ($\epsilon_{Nd}$) and selected Post-Archean Australian Shale (PAAS) normalized Rare Earth Elements (REEs) ratios of different external sources for the South Pacific Gyre**

| External sources | $\epsilon_{Nd}$ | PAAS normalized Ce anomaly (Ce/Ce*$_{PAAS}$) 2×Ce$_{PAAS}$/(La$_{PAAS}$+Pr$_{PAAS}$) | PAAS normalized heavy REE enrichment relative to light REE (HREE/LREE$_{PAAS}$) (Yb$_{PAAS}$+Lu$_{PAAS}$)/(Pr$_{PAAS}$+Nd$_{PAAS}$) |
|---|---|---|---|
| Volcanogenic material (represented by the pumice) | +7.5 ± 0.2 | 0.89 ± 0.02 | 7.06 ± 0.47 |
| Australian dust | −29 to +2[43] | 0.92 to 1[81,82] | 0.94 to 1.15[81,82] |
| Basaltic volcanic islands | 0 to +10[39] | 0.97[40] | 0.65[40] |

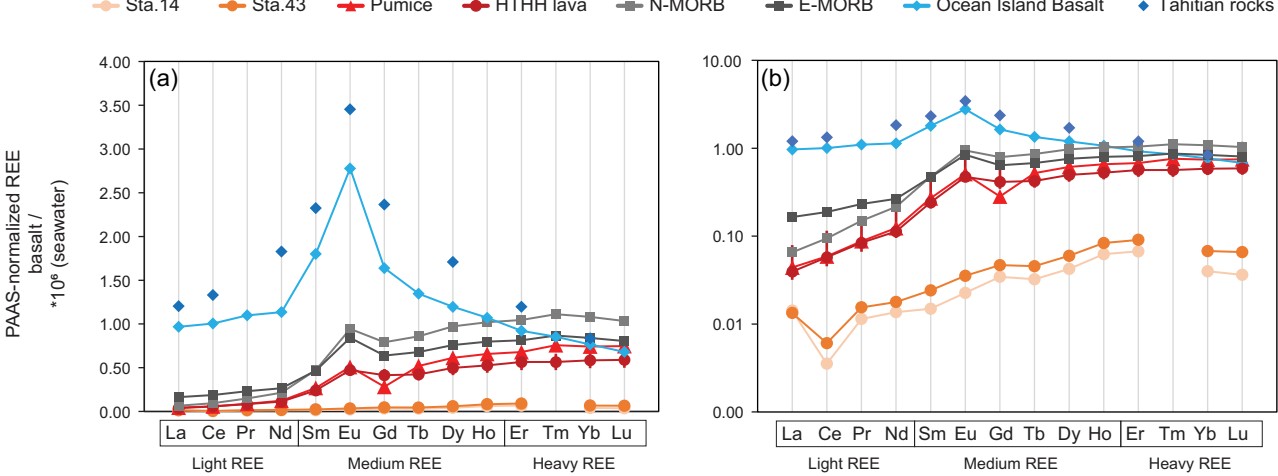

**Fig. 4 | A comparison of the Post-Archean Australian Shale (PAAS) normalized Rare Earth Element (REE) patterns in surface seawater (uppermost 3-5 m) and various types of basalts.** Subplots (**a**) and (**b**) are identical, with the y-axis in (**a**) presented in a linear scale, while the y-axis in (**b**) employs a logarithmic scale. Surface seawater REE data from stations 14 and 43 were chosen to represent the samples from the eastern and western South Pacific Gyre, respectively. REE data for mid-ocean ridge basalts (MORB) and ocean island basalts (OIB) are from Sun and McDonough[40], where those values are based on a literature review and internal consistency of elemental ratios. N-MORB represents normal MORB and E-MORB represents enriched MORB. REE data for Hunga Tonga-Hunga Ha'apai (HTHH) lava are from Ewart et al. [42] where the error bar represents the standard deviation of five lava samples. REE data for Tahitian rocks are from Cordier et al. [83] which is shown as a representative of volcanic island basalt in the South Pacific. The REE pattern of these basalts normalized to Primitive Mantle is available in Supplementary Fig. 5.

environment and magma generation processes[40]. These factors lead to systematic compositional differences between MORBs and OIBs. Therefore, the Primitive Mantle-normalized REE pattern of the pumice closely resembled that of normal MORBs[40], and differed from that of OIBs[40] (Supplementary Fig. 5). In addition, basalts formed within island arcs exhibit geochemical diversity as a function of different mantle origins and/or varying melting processes, which result in distinguishable elemental compositions[41]. Lavas from different Tonga-Kermadec islands display distinctive REE patterns[42], and our collected pumice has a Primitive Mantle-normalized REE pattern similar to that of the lava from the HTHH volcano (Supplementary Figs. 5, 6). Therefore, the geochemical composition of our collected pumice strongly suggests that it originated from the HTHH eruption.

When the REE concentrations were normalized to PAAS, the pumice and HTHH lava exhibited a more pronounced enrichment in HREE (Fig. 4a) and no Ce anomaly compared to surface seawater (Fig. 4b). This is consistent with the REE characteristics of the external source to the western SPG and explains the correlated changes in Ce anomaly and HREE enrichment in the surface waters between 90°W and 170°E (Fig. 3). OIBs exhibit a pronounced enrichment of MREEs, a crustal Ce signal (without anomaly), and little fractionation between light and heavy REEs (i.e. no HREE enrichment) (Fig. 4). Volcanic island weathering imprints an OIB-like REE pattern in local fresh or brackish waters at the island-ocean interface, although this signal is typically rapidly lost in coastal seawaters due to scavenging[25,28,38]. Overall, input of volcanic island weathering cannot account for the observed changes in REE ratios of surface seawater in the SPG (Fig. 3). Therefore, it

can be concluded that volcanic island weathering was not the source of the material driving the observed geochemical changes.

Another possible external source to the western SPG is Australian dust[43,44]. However, the $\epsilon_{Nd}$ signatures of the Australian dust (−29 to +2)[43] are generally too unradiogenic to account for the observed radiogenic signature (+1) near the Kermadec arc. In addition, Australian dust has an overall shale-like REE pattern lacking a Ce anomaly and a HREE enrichment in contrast to surface seawater (Fig. 3). Furthermore, our western part of the transect is located at a distance from the region receiving long-range dust transport from Australia, which follows a southeastward trajectory[44]. Therefore, it is unlikely that Australian dust is a major component of the external source.

Shallow hydrothermal vents (<500 m depth) along the Tonga-Kermadec arc potentially increase Fe concentrations in the photic layer through vertical transport[45]. Although dissolved $\epsilon_{Nd}$ may show a slightly more radiogenic signature within hydrothermal plumes (one $\epsilon_{Nd}$ unit shift)[46], REEs are typically immediately immobilized near the vents upon contact between hot fluids and cold bottom waters[47]. Furthermore, no significant excess of $^3$helium ($^3$He$_{xs}$), a conservative tracer for hydrothermal plumes[48], was detected at our station above the Monowai Seamount on the Kermadec ridge. The measured $^3$He levels were even lower than the background $^3$He value (~ 2.38 fmol/kg) (Fig. 5c). In addition, hydrothermal fluids commonly exhibit a pronounced, positive europium (Eu) anomaly (Eu/Eu* > 1)[49]. The absence of a positive Eu anomaly in the water column near the Tonga-Kermadec arc (Fig. 5b) suggests a negligible shallow water hydrothermal input. Moreover, $\epsilon_{Nd}$ signatures become more radiogenic

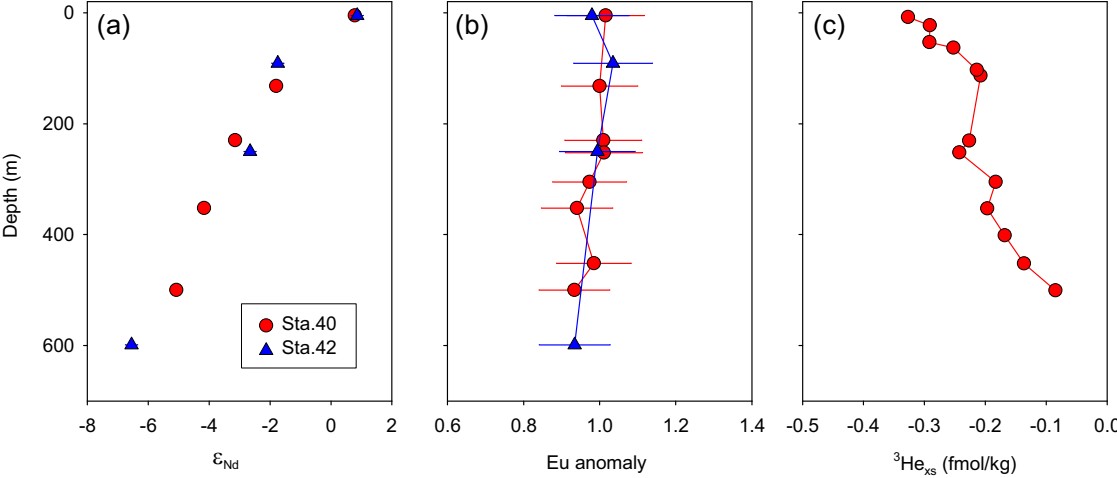

**Fig. 5 | Vertical distribution of (a) Nd isotopic composition ($\varepsilon_{Nd}$), (b) Post-Archean Australian Shale (PAAS) normalized Eu anomaly, and (c) excess $^3$He ($^3$He$_{xs}$) in the upper 600 m at stations close to the Tonga-Kermadec arc.** Station 40 is on the flank of the Monowai seamount. The external 2 SD of the $\varepsilon_{Nd}$ is smaller than symbol size. PAAS normalized Eu anomaly (Eu/Eu*$_{PAAS}$) is calculated as 2*Eu$_{PAAS}$/(Gd$_{PAAS}$ +Sm$_{PAAS}$). Error bars for Eu anomaly are derived from the propagation of measurement errors (2 SD) of each element. Helium (He) isotope data are not available for station 42 in the upper 600 m.

from depths of 600 m towards the surface (Fig. 5a), suggesting external inputs into surface seawaters from above rather than vertical supply from below.

In summary, we conclude that the dissolution of volcanogenic material originating from the HTHH eruption, which occurred 39 to 79 days prior to our sampling, is predominantly responsible for the radiogenic $\varepsilon_{Nd}$ signature, weak Ce anomaly and strong heavy REE enrichment observed in the western SPG along GP21. The area of the strongest geochemical signal largely overlaps with the main region of post-eruption volcanic ash deposition as well as the location of floating pumice (Fig. 2; Supplementary Fig. 3). In addition, surface currents may redistribute the volcanic input signal up to 600-2400 km away in 70 days. The gradual change in $\varepsilon_{Nd}$ signatures and REE ratios between the western and eastern SPG may be indicative of the eastward extension of the influence of the volcanic eruption through atmospheric ash dispersion (Fig. 1c; Supplementary Fig. 4) and/or the predominant direction of the surface ocean currents (Fig. 1a).

### Rapid release and scavenging of Nd following the volcanic input
The SPG is known for the lowest surface water REE concentrations in the global ocean, likely due to its remoteness with a limited amount of terrestrial inputs and a lack of vertical supply due to strong water column stratification[50]. Despite the clear evidence for partial dissolution of volcanogenic materials discussed above, surface water Nd concentrations in the western SPG remained consistently low (~4 pmol/kg) at levels comparable to previous measurements[36,37] (Fig. 2b). Assuming an initial surface seawater Nd concentration of ~4 pmol/kg with an $\varepsilon_{Nd}$ value ~−2 in the western SPG, an addition of 0.9-1.9 pmol/kg (of seawater) of Nd from the volcanogenic material with an $\varepsilon_{Nd}$ value of +7.5 is required to obtain a surface $\varepsilon_{Nd}$ value of ~0 to +1. In the absence of subsequent scavenging from the surface water[51], this would result in a 22−46% increase in Nd concentration. Given that the samples were collected 9-10 weeks after the eruption, subsequent particle scavenging was likely to have occurred, which had offset the increase in concentration. The preferential removal of Nd and other LREEs during particle scavenging, following the dissolution of volcanic material, resulted in a downward shift in the Ce/Ce* to HREE/LREE slope. This explains the deviation of the linear regression between Ce anomaly and HREE enrichment from the pumice endmember (Fig. 3).

Surface layer Nd residence times were suggested to range from 1–4 years in the eastern Indian Ocean and eastern Atlantic Ocean[39,52]. However, the mean residence time of Nd in the mixed layer is dependent on various factors such as REE inventory, supply rate or export flux, and particle density[39,53]. Trace metals are shown to exhibit a markedly shorter residence time in response to large dust deposition events due to enhanced particle scavenging[54–56]. Similarly, ash particles have residence times ranging from hours to days in the mixed layer, depending on their particle sizes[21]. Following ash deposition, the rapid downward flux of these particles may facilitate rapid Nd scavenging, resulting in a short residence time of REE in the mixed layer. This enables the surface water REE pattern to be rapidly altered in response to local external sources (Fig. 3).

### Trace metal fertilization and biological response in the SPG
The concentrations of surface dissolved Al (dAl), Mn (dMn) and Fe (dFe) all increased between the eastern and western SPG (Fig. 6 a,b), accompanying the overall increase in $\varepsilon_{Nd}$ values (Fig. 2a). Previous studies have attributed elevated dFe concentrations in the western South Pacific to (i) shallow inputs of Fe of hydrothermal origin from the vents along the Tonga-Kermadec arc[45,57], or (ii) direct interaction of seawater with the islands or Australian continent, which can subsequently be advected elsewhere by currents[10,58]. However, as discussed above, volcanic material was the dominant external source of trace metals in the western SPG during our cruise, with a greater influence than other external sources. In addition, a comparison of surface trace metal concentrations in the western SPG between our western transect and other two earlier GEOTRACES cruises, GP13 and GP19, revealed elevated levels in the region 175°E-175°W for our cruise, with a notable increase in dAl over an extended area (Supplementary Fig. 7). It must be acknowledged that the temporal gap between our sampling and the eruption (9–10 weeks) requires consideration. For instance, approximately two to three months after the 2010 Eyjafjallajökull eruption, Achterberg et al.[21] observed no elevated surface dFe in offshore waters of the Iceland Basin. This midsized eruption, however, emitted tephra in an amount (~3.8 × 10$^{14}$ g) that was one magnitude lower than the major HTHH eruption (~2.9 × 10$^{15}$ g)[14]. Consequently, the HTHH eruption appeared to exert a stronger and more far-reaching geochemical impact in comparison to those previously observed for midsized eruptions.

In the SPG, the dAl closely followed the change in surface $\varepsilon_{Nd}$, showing a highly significant positive correlation ($n = 19$, $R^2 = 0.88$, $p < 0.001$, Fig. 6c). This suggests that dAl had the same source as the radiogenic $\varepsilon_{Nd}$, that is, the volcanic aluminosilicate input. However, there was a slight deviation in the increase exhibited by dMn compared

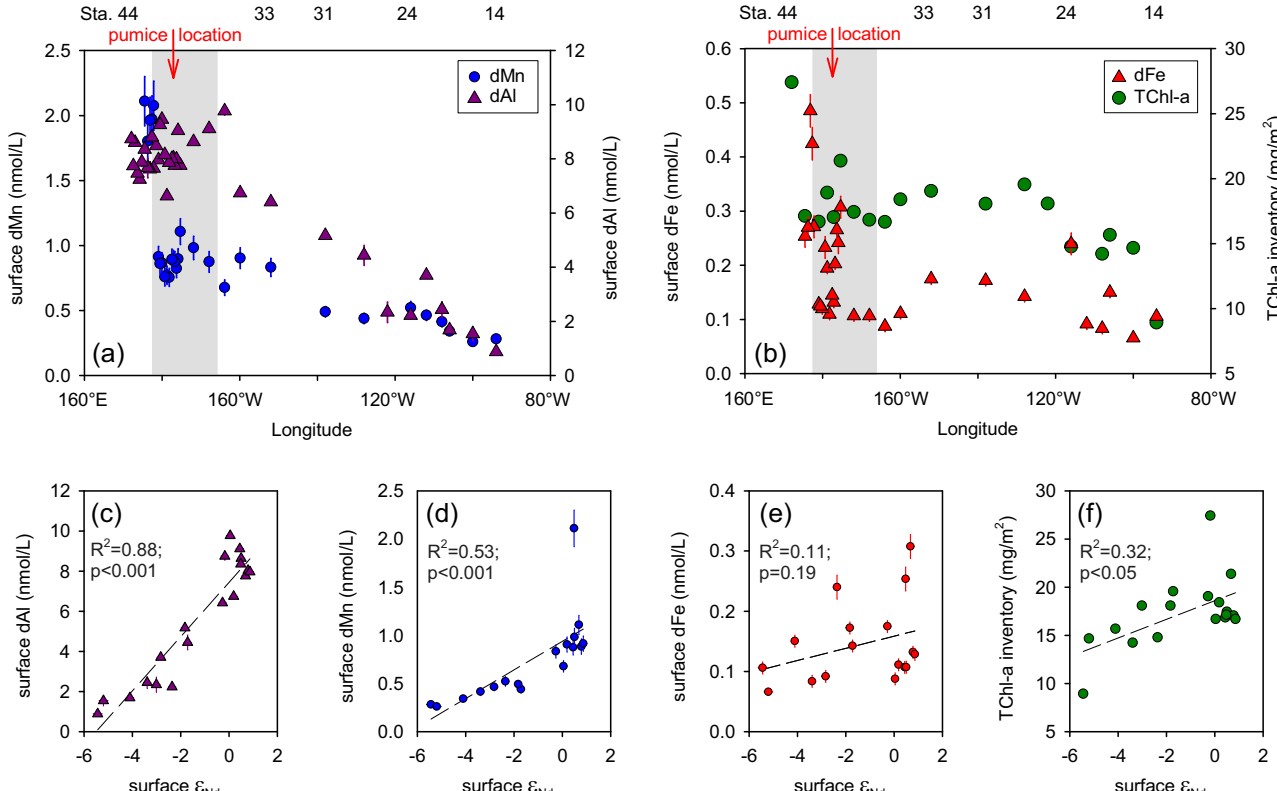

**Fig. 6 | Changes in surface water (uppermost 3-5 m) trace metal concentrations and total chlorophyll a (TChl-a) inventory in the euphotic layer of the South Pacific Gyre (170°E-90°W). a** Surface dissolved manganese (dMn) and aluminum (dAl) concentrations; **b** surface dissolved iron (dFe) concentration and TChl-a inventory in the euphotic layer; **c–f** cross plots of surface dAl, dMn, dFe, and TChl-a inventory versus Nd isotopic composition ($\varepsilon_{Nd}$). The location of the pumice is indicated by a red arrow pointing downwards. The region of ash deposition within 3 days of the eruption, based on the NOAA HYSPLIT volcanic ash dispersion model (Fig. 1b, Supplementary Fig. 3), is shown in the transparent gray box. The external 2 SD of the $\varepsilon_{Nd}$ values are smaller than the symbol size. The error bars of dAl, dMn, and dFe represent 1 SD of repeated measurements of these trace metals, respectively.

to the near linear increase observed by dAl (Fig. 6a). This is documented by its weaker positive correlation with surface $\varepsilon_{Nd}$ (n = 18, $R^2$ = 0.53, p < 0.001, Fig. 6d). In contrast, the concentrations of dFe decreased to very low levels (-0.1 nmol/L) in the main ash deposition area (shaded in Fig. 6b). As a result, we observe insignificant correlations between surface $\varepsilon_{Nd}$ and dFe (n = 18, $R^2$ = 0.11, p = 0.19, Fig. 6e).

Given the time gap of 9–10 weeks between the HTHH eruption and our observation in the western SPG, the observed differences between dAl, dFe, and dMn can be attributed to their scavenging behavior and biological requirements, which result in a markedly shorter residence time for dFe compared to dAl and dMn[59]. In open ocean surface waters, dAl distributions are primarily controlled by aeolian inputs, as it is not known to be actively assimilated by organisms, while dMn is essential for organisms and surface ocean concentrations are further buffered by photochemical reduction[60,61]. In contrast, dFe is scavenged from seawater more rapidly than dAl or dMn and has a greater requirement for phytoplankton growth than dMn[61,62]. Dust-seeding mesocosm experiments have indicated that a large dust deposition event can accelerate the export of Fe from the water column through scavenging, while a remarkable increase of dissolved Al and Mn was observed[56]. Extremely short residence times, with timescales of days to weeks, were reported for Fe in Atlantic surface waters receiving enhanced atmospheric inputs from Saharan origins[54,55]. Consequently, the more pronounced depletion of dFe in the main ash-deposition area (Fig. 6b), despite the anticipated high input fluxes, is primarily a result of scavenging subsequent to the release and/or biological removal[21].

A weak positive correlation was observed between surface $\varepsilon_{Nd}$ and the total chlorophyll-a (TChl-a) inventory in the euphotic layer

measured on the research cruise (n = 18, $R^2$ = 0.32; p < 0.05, Fig. 6f). In addition, a positive satellite-derived Chl-a anomaly for March 2022 was evident in the western part of the transect compared to the March average for the period between 2002 and 2023 (Supplementary Fig. 8). Together these indicate a potential increase in phytoplankton growth in response to the volcanic input of dFe via stimulation of $N_2$ fixation, particularly in the western SPG, as shown in Fig. 6b. It should be noted, however, that the time interval after the eruption may have precluded the observation of the most elevated phytoplankton biomass in the western SPG. Given that multiple sources of trace metals are known in the western SPG, confidently ascribing this elevated phytoplankton biomass to volcanic dFe fertilization over and above other sources is challenging. However, as discussed above, volcanic input was identified as the dominant TM source to surface waters, therefore we consider a linkage between volcanic input from the HTHH eruption into the western SPG and the enhanced TChl-a inventory a plausible scenario.

**Constraints on volcanic Nd and Fe supply to the SPG**

Extrapolating the Nd concentration in the pumice (4.18 μg/g) to the total ejecta of the HTHH eruption ($2.9 \times 10^{15}$ g)[14], the ejecta contain a total of $1.2 \times 10^{10}$ g of Nd. The NOAA HYSPLIT volcanic ash dispersion model indicated a mass loading area coverage of $5.4 \times 10^6$ km$^2$ three days after the eruption (Fig. 1b), which can be considered to represent the lower boundary of the ash depositional area. Meanwhile, our data demonstrate a significant increase of 2–3 $\varepsilon_{Nd}$ units between stations 33 and 44 (152°W to 172°E) in the western SPG. Based on this longitudinal coverage and the latitudinal range of 30° (20°S to 50°S) indicated by the HYSPLIT model (Fig. 1b), we can estimate the upper limit of the

depositional area to be $1.2 \times 10^7 \, km^2$. On the basis of an estimated overall release of 0.9–1.9 pmol/kg Nd within the surface mixed layer (~50 m), we calculate a total release of $0.4–1.6 \times 10^8$ g of Nd. The upper limit is comparable to, although slightly lower than, the annual global dust input of Nd ($2–4 \times 10^8$ g/yr)[63,64]. It represents 1.3 % of the total Nd contained in the ejecta from HTHH, which is lower than the estimated fraction of soluble Nd in dust (2–50 %)[64]. Although Nd scavenging is clearly evident based on the lack of a Nd concentration enrichment in surface waters of the depositional area, this release still had a significant impact on the Nd isotope composition and REE characteristics of surface waters, making them sensitive tracers of volcanic activity in the present and past ocean.

Volcanic ash also rapidly supplies significant amounts of soluble Fe to seawater, with a global mean estimate of $200 \pm 50$ nmol Fe/g ash for subduction zone volcanic ash[3]. Multiplying this mean value by the total amount of ejecta from the HTHH eruption ($2.9 \times 10^{15}$ g)[14] results in a potential release of $5.8 \pm 1.5 \times 10^8$ mol ($3.2 \pm 0.8 \times 10^{10}$ g) of Fe. This estimate likely represents an upper limit, as some types of ejecta, such as pumice, release Fe less efficiently than volcanic ash upon contact with seawater[65]. However, the long-lived buoyancy of pumice potentially promotes a continuous release of Fe into surface waters for months to years before finally sinking[22,23]. The long-term Fe release efficiency of pumice to surface seawater should be further investigated to test this possibility. This estimate is comparable to the Fe fertilization amount from the similar giant eruption of Mount Pinatubo in June 1991 (VEI 6; $7.2 \times 10^8$ Fe), which was associated with atmospheric carbon/oxygen anomalies[7], but it is much larger than the estimated Fe release from previous midsized eruptions, including the Kasatochi eruption in August 2008 (VEI 4; ~ $1 \times 10^8$ mol Fe), which stimulated a massive phytoplankton bloom[66], and the Eyjafjallajökull eruption in April-May 2010 (VEI 4; $1.8 \times 10^6$ mol Fe), which resulted in enhanced major nutrient drawdown[21].

The New Zealand – Tonga – Kermadec arc is estimated to emit $20–26 \times 10^{12}$ g/yr volcanic ash[3], corresponding to $30–78 \times 10^5$ mol/yr Fe, suggesting that Fe release from HTHH approximates the integrated Fe contribution of the New Zealand – Tonga – Kermadec arc volcanic input over a century. Pavia et al.[24]. estimated a dust-borne Fe flux to the central region of the SPG at 3.5–11 µmol/m²/yr. Multiplying this estimate with the surface area of the SPG (37 million km²) yields an estimated dust-borne Fe flux to the SPG of $1.3–4.1 \times 10^8$ mol/yr. Consequently, the quantity of Fe released from the HTHH eruption is equivalent to the annual dust-borne Fe flux into the entire SPG. In order to improve the accuracy of biogeochemical models in the Pacific Ocean, it is, therefore, necessary to incorporate the episodic volcanic input of trace metals and to improve parameterization of the key chemical-biological processes that respond to volcanic inputs in the models.

## Methods
### Volcanic ash dispersion model and trajectory model
The NOAA HYSPLIT volcanic ash dispersion model[67] was used to calculate the transport and dispersion of volcanic ash from the Hunga_Tonga-Hunga_Ha'apai (HTHH) volcanic eruption. Information about the HTHH volcano was extracted from the Smithsonian Institution "Volcanoes of the World" database. Archived GDAS (1 degree, global, 2006-present) data was used for meteorology. Eruption source parameters (ESPs) for the HTHH volcanic eruption, as assigned by the United States Geological Survey (USGS), were used for the mass eruption rate. The HYSPLIT model was run for 72 hours with 3 concentration layers after the eruption (15th January 2022, 09:00 UTC) with a 24 hours output interval. Model output displays results of one-hour averages at the given snapshot interval for 3 altitude layers: 6000 m, 12000 m, and 18000 m.

The NOAA HYSPLIT trajectory model[67] was used to calculate forward trajectories after the HTHH eruption (15th January 2022, 09:00

UTC) from the eruption location. A maximum of 315 hours can be specified for archive datasets on the website. Archived GDAS (1 degree, global, 2006-present) data was used for meteorology. "Model vertical velocity" that uses the vertical velocity field from meteorological data was chosen for the type of vertical motion method the trajectory model uses in its calculation.

### Radiogenic neodymium isotopes sampling and analyses
A total of 28 surface seawater samples were collected for the radiogenic Nd isotope measurements. Approximately 20 L of surface seawater was collected for stations 1-14 using 10 L Niskin bottles attached to a stainless steel CTD rosette. Starting from station 16, ~40 L of surface seawater was collected per sample from a towed, trace-metal-clean near-surface seawater sampling device equipped with acid-washed PVC tubing, and pumping provided by a Teflon diaphragm pump (i.e. tow-fish), either immediately before or after the station, to allow isotopic composition measurements at anticipated low concentrations. Between stations 39 and 44, an additional 17 surface samples (1-2 L) were collected from the tow-fish for REE measurements.

The samples were treated onboard strictly following recommended GEOTRACES protocols[68]. Each 20 or 40 L sample was filtered through a nitro-cellulose acetate filter (0.45 µm pore diameter) into one or two acid-cleaned LDPE-cubitainers (20 L) with a peristaltic pump within 2 h after collection, and subsequently acidified to pH-1.9 with concentrated, distilled HCl. For the determination of Nd (using isotope dilution) and REE concentrations, 1-2 L aliquots from each filtered sample were collected in acid-cleaned 1 or 2 L PE-bottles and acidified to pH-1.9. To each large volume sample, 25 µL of $FeCl_3$ solution (~200 mg Fe/mL) was added per 1 L of sample, and the sample was left to equilibrate for 24 h. Ammonia solution (25%, Merck Suprapur®) was next added to increase the pH to 7.8-8.2. After 48 h the trace elements co-precipitated with the iron hydroxide precipitates and settled to the bottom of the cubitainers, and the supernatant was syphoned off. The precipitates were then transferred into 2 L PE-bottles and transported to GEOMAR for analysis.

Precipitates were dissolved and purified for measurement with Multicollector-Inductively Coupled Plasma Mass Spectrometer (MC-ICP-MS) in the home laboratory at GEOMAR. Detailed purification procedures can be found in Rahlf et al.[69]. Briefly, the precipitates were dissolved in 6 M HCl/ 0.5 M HF and dried down, and then treated with aqua regia at 120 °C for 24 h to remove organic compounds. Then pre-cleaned di-ethyl ether was used to remove most Fe via liquid–liquid extraction. Neodymium was chromatographically separated from matrix elements using first a cation exchange resin AG 50W-X8 (1.4 ml, 200–400µm) and then from the other REE using Eichrom®LN-Spec resin (2 ml, 50–100 µm) on a second chromatographic column. The $^{143}Nd/^{144}Nd$ ratios were determined using a Neptune Plus MC-ICP-MS at GEOMAR and corrected for instrumental mass bias to $^{146}Nd/^{144}Nd = 0.7219$ and to $^{142}Nd/^{144}Nd = 1.141876$[70]. The $^{143}Nd/^{144}Nd$ ratios of all samples were normalized using bracketing analyses of the JNdi-1 standard, with a value of 0.512115[71]. The total procedural laboratory blanks for water samples (n = 3) were negligible at <20 pg for Nd and contributed a maximum of 0.3 % to the seawater samples. $\varepsilon_{Nd}$ is defined by the equation:

$$\varepsilon_{Nd} = \left( \frac{(^{143}Nd/^{144}Nd)_{sample}}{(^{143}Nd/^{144}Nd)_{CHUR}} - 1 \right) \times 10^4 \qquad (1)$$

where the $^{143}Nd/^{144}Nd$ of CHUR (Chondritic Uniform Reservoir) are 0.512639[72].

The external reproducibility (2 × standard deviations, 2 SD) of the Nd isotope measurements was assessed by repeated purification and measurement of USGS reference material NOD-A-1 at concentrations matched to those of the samples (12-20 ppb, $\varepsilon_{Nd} = -9.35 \pm 0.15$, n = 7 for

repeated purification, n = 35 for repeated measurement), which is used to demonstrate the error of the measured $\varepsilon_{Nd}$ in all figures (2 SD = 0.15). If the internal error (2 × standard errors, 2SE) of a sample was larger than the external error, the internal error is used.

Pumice was rinsed with Milli-Q water (Millipore) three times and then dried and homogenized prior to alkaline fusion following Bayon et al.[73]. Neodymium in the digestion solution was then purified by chromatography following the same procedure as for seawater. The accuracy and reproducibility of the fusion technique and measurement were monitored by processing reference materials with each batch of samples including USGS reference material BHVO-2 ($\varepsilon_{Nd}$ = +6.80 ± 0.24, $n$ = 3) and AGV-2 ($\varepsilon_{Nd}$ = +2.99 ± 0.13, n = 3). The larger 2 SD from BHVO-2 measurement is used to demonstrate the error of the measured $\varepsilon_{Nd}$ in the pumice (2 SD = 0.24).

### Rare earth element concentration analyses

For the precise determination of Nd concentrations, 1 L samples were spiked with a pre-weighed $^{150}Nd$ spike, after which the Nd was precipitated with $FeCl_3$ solution and purified with cation exchange resin AG 50W-X8 (1.4 ml, 200–400 μm). The Nd concentration was then measured on the Neptune Plus MC-ICPMS via the isotope dilution (ID) method based on $^{150}Nd/^{144}Nd$. External reproducibility (2 SD) was better than 0.4 % for Nd according to repeated treatment and measurement of the same sample (n = 4).

The rare earth elements (REEs) were pre-concentrated offline using a SeaFAST system (model M5 from Elemental Scientific) which employs a NOBIAS PA-1 resin column to preconcentrate the REEs and other trace metals[74]. A 10 ng/g thulium solution (25 μL) was added to each 25 mL acidified seawater sample prior to preconcentration to monitor yields. The same was done with acidified pure Milli-Q water blanks, and reference seawaters (pH-1.9) which were preconcentrated like the samples. Using the M5 SeaFast system, 24 mL of sample was precisely loaded onto the column, the matrix was washed away, and then the REEs were eluted with 400 μL of 1.5 M $HNO_3$. Before measurement, 200 μL of 10 ng/g Re in 0.1% $HNO_3$ were added to each sample as an internal standard and to account for any evaporation of the samples in the meantime. All samples were analyzed on a Thermo Element XR ICP-MS coupled with a CETAC "Aridus 2" desolvating nebulizer to reduce oxide formation. Oxide formation was monitored at the start of each analytical session using single element solutions[74] and was negligible at <0.05%.

GEOTRACES inter-calibration seawater samples BATS 15 m and 2000 m[75], as well as seawater samples from this cruise (Station 36, 5 m and 5289 m), were repeatedly measured to monitor external reproducibility and accuracy. In this work, the 2 SD of repeated measurements of the surface seawater at station 36 (n = 6) is used to represent the uncertainties of REEs of all the surface samples, which is <10% for all REEs, except for Ce (14.9 %). The average total procedural onboard blank (n = 3) was ≤3.54 % for all REEs, except for Ce (8.59 %), compared to the surface sample containing the lowest REE concentrations along the transect. Blanks, mean values and 2 SD for the reference seawaters are provided in Supplementary Data 1. A comparison of Nd concentrations measured by SeaFast and by the isotope dilution method in this study shows consistency between the two methods for Nd (Nd-SeaFast = 1.01 × Nd-ID, $R^2$ = 0.99; Supplementary Fig. 9).

The REE concentrations of pumice digestion solution was measured on an Agilent 7500cx Quadrupole-ICP-MS at GEOMAR. Oxide formation was corrected by measuring element solutions of barium (Ba), Ce, praseodymium (Pr) + Nd, and samarium (Sm) + europium (Eu) + gadolinium (Gd) + terbium (Tb) at the start of the analytical session. The reproducibility was monitored by repeated measurements of USGS reference material BHVO-2. Mean values and 2 SD for the BHVO-2 measurements are given in Supplementary Data 1.

### Surface dissolved manganese (dMn), dissolved iron (dFe), and dissolved aluminum (dAl)

Surface seawater for trace metal measurements was collected from the trace-metal-clean underway tow-fish and filtered through a 0.8/0.2 μm filter cartridge (AcroPak 500, Pall) into pre-cleaned 125 mL LDPE bottles (Nalgene), then acidified with hydrochloric acid (Ultrapure, Romil) for storage and stabilization (pH ~1.9). All sample and standard preparations were completed under a laminar flow bench, and laboratory and analytical work was undertaken in an ISO Class 5 clean lab. All LDPE and FEP bottles used for reagents, standards, and sample preparation were acid washed according to GEOTRACES protocols[68].

Dissolved Mn and Fe analytes were initially pre-concentrated and buffered online to pH 6.4 using SeaFast (SC-4 DX; ESI), then measured on the ICP-MS (Element-XR, ThermoFisher) at GEOMAR. The ammonium acetate buffer was prepared using Optima grade (Fisher) glacial acetic acid and ammonium hydroxide. Subboiled distilled nitric acid was used to make up the 1 M elution acid, with a 250 ng $L^{-1}$ indium spike added to manage any potential drift correction. Additional details about this methodology are reported in Rapp et al.[76]. Quantification and accuracy of trace metals was assessed using standard addition ($R^2$ = 0.99) as well as certified reference materials (SAFe D2; Cass-6 and Nass-7, National Research Council Canada), achieved to good agreement with reported consensus values (weighted mean absolute percentage error (WMAPE): <5.5% for dFe and <11% for dMn). Several seawater samples were run as duplicates or triplicates and returned relative standard deviations (RSDs) of <6.7% (dFe) and <9% (dMn), which are used to represent the error of each sample. The limit of detection was 0.005 nmol/L for dMn, and 0.05 nmol/L for dFe.

Dissolved Al was analyzed shipboard following the methodology of Ren et al.[77]. Briefly, a working reagent consisting of 40 mg/L lumigallion and 2 M ammonium acetate (pH ~6) was prepared and stored in dark at the fridge. Working reagent (250 μL) was added into the 5 mL acidified seawater (pH ~1.9), resulting in a final pH of 5.0-5.5. Samples were then placed in an oven at 80 °C for 3 h to accelerate Al-lumogallion complexation. An eight-point calibration line (0, 1, 2, 4, 8, 12, 15, 20 nmol/L Al standard additions to surface seawater from South Atlantic Gyre), blank determination and an internal reference sample were pretreated in the same manner as samples and measured daily. All samples were analyzed using a Cary Eclipse fluorimeter; emission and excitation wavelengths were set to 507 − 575 nm, with a 10 nm slit width. Blank contributions were determined as two separate parts, the manifold blank and the reagent blank. The manifold blank was assessed by the average counts of two acidified seawater samples (one surface and one deep water) without reagents. The reagent blank was determined using two different methods: (1) by analyzing three acidified seawater samples spiked with 1x, 2x, and 3x reagent volume and (2) using two calibration series with 1x and 2x reagent volume, respectively. The reagent blank was then assessed from the difference in the intercepts of two series of calibration lines. For each seawater sample, duplicates or triplicates were performed, and the error of each sample is reported as the 1 SD of repeated measurements.

### Total chlorophyll-a (TChl-a) concentrations

Seawater samples for phytoplankton pigment analysis using high-performance liquid chromatography (HPLC) were collected in 10 L opaque carboys at six depths throughout the euphotic zone. The depths were identified by the photosynthetically active radiation (PAR) intensity using a PAR sensor and an in vivo fluorescence sensor attached to the CTD (i.e., 100%, 50%, 25%, 10%, 1%, 0.1% surface PAR). Samples (4 L) were filtered onto 25 mm diameter Fisher MF300 GF/F filters and frozen immediately at −80 °C. Upon return to land, pigments were extracted in 90% acetone in plastic vials by homogenization of the filters using glass beads in a cell mill, centrifuged (10 minutes, 4500 g, 4 °C), then the supernatant was filtered through 0.2 μm polytetrafluoroethylene filters (VMR International) and

subsequently quantified by reverse-phase HPLC (Dionex UltiMate 3000 LC system, Thermo Scientific)[78]. Pigment standards were from Sigma-Aldrich (USA) and the International Agency for 14 C Determination (Denmark). The total chlorophyll-a (TChl-a) concentration was calculated as the sum of chlorophyll-a and divinyl chlorophyll-a concentrations. The TChl-a inventory in the euphotic zone was calculated through the trapezoidal integration method.

### Helium (He) isotopes

Seawater samples for He isotopes were collected using a copper pipe connected to the Niskin bottle, with water flowing until all bubbles in the tube were purged. Subsequently, the pipe was securely sealed using an electrical drill and a ratchet to ensure a tight seal. Additionally, 0.5 L samples for tritium analyses were collected at full depth at Station 40 to account for tritium concentrations in the deep water, thus correcting the helium isotope measurements. The helium isotopes and tritium samples were analyzed at the University of Bremen following the procedures described by Sültenfuß et al.[79].

The helium $^3He/^4He$ isotope ratio (R), is expressed relative to the atmospheric ratio ($R_a = 1.38 \times 10^{-6}$), using the delta notation:

$$\delta^3He\,(\%) = \left(\frac{R}{R_a} - 1\right) \times 100 \qquad (2)$$

The background $^3He$ concentration (-2.38 fM) in the deep ocean was calculated from the background $\delta^3He$ (~ −1.7 %) and equilibrium He concentrations (-1.75 nM) at the observed in situ salinity (~35 PSU) and temperature (~3.5 °C). The excess of $^3He$ ($^3He_{xs}$) was then calculated by subtracting the background $^3He$ and $^3H$ levels from the measured $^3He$ concentrations.

### Reporting summary

Further information on research design is available in the Nature Portfolio Reporting Summary linked to this article.

## Data availability

Maps and figures were generated using Adobe Illustrator, Ocean Data View[80], and Sigmaplot. All the data generated in this study are provided in the Supplementary Data file. All the data generated in this study have been deposited in the PANGAEA database (https://doi.org/10.1594/PANGAEA.971875) and Figshare repository (https://doi.org/10.6084/m9.figshare.26389948).

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

## Acknowledgements
We gratefully acknowledge funding for RV SONNE ship-time (SO289) by the German Federal Ministry of Education and Research (BMBF) (Förderkennzeichen 03G0289NA). We thank the captain and crew of RV Sonne for their help and support during cruise SO289. We are highly grateful to Xue-gang Chen for helping to process the Helium isotope data. We also thank Tabea von Keitz, Nikoleta Vitsou, Nijat Adigozalli and Sieglinde Kolbrink for laboratory support. Zuozhu Wen and Julianne Tammen are thanked for assistance at sea. We gratefully acknowledge the NOAA Air Resources Laboratory (ARL) for the provision of the HYSPLIT transport and dispersion model and/or READY website (https://www.ready.noaa.gov) used in this publication. We also acknowledge the NASA Ocean Biology Processing Group (OBPG) and the Ocean Biology Distributed Active Archive Center (OB.DAAC) for providing the data and support used to create the map of satellite-derived chlorophyll-a anomalies in this work (https://oceancolor.gsfc.nasa.gov/l3/order/). Z.Z. and K.G. acknowledge funding support from the BMBF (Förderkennzeichen 03G0289NA).

## Author contributions
Z.Z. and M.F. designed and coordinated the research; M.F. and E.A. acquired and managed the funding; Z.Z. carried out the sampling and analytical work with the help from A.X., E.H., M.G. and Z.S.; K.G. provided dFe and dMn data; T.L. provided dAl data; T.B., Z.Y., and H.L. provided TChl-a data; R.K. contributed to the interpretation of data; Z.Z. wrote the manuscript, and all co-authors reviewed and contributed to the final version.

## Funding

## Competing interests
The authors declare no competing interests.
