## [Transparent Peer Review file · Nature Communications]

Substantial trace metal input from the 2022 Hunga Tonga-Hunga Ha'apai eruption into the South Pacific

Corresponding Author: Dr Zhouling Zhang

Version 0:

Reviewer comments:

Reviewer #1

(Remarks to the Author)

I enjoyed reading the manuscript by Zhang et al. on trace metal input from the 2022 Hunga Tonga-Hunga Ha'apai eruption into the South Pacific Gyre. The dataset is impressive, experimental design was strong, and the manuscript was well-written. I recommend moderate revisions but feel that the paper is publishable in Nature Communications – and indeed that the authors have some connections to draw at the end of the paper to make its impact even higher.

The difference in eNd and trace metals before and after the eruption was striking. I feel convinced that indeed, this eruption altered the REE and trace metal budgets of the surface of the SPG.

I had one most significant issue reading the paper. It's discussed in lines 218-228 and Figure 3. It seems like there's a problem with the authors' proposed hypothesis of scavenging followed by volcanic material dissolution, because the trend of the data doesn't lead towards either the HTHH lava or the pumice endmember. The authors did not really explain why this is the case, or if they did I didn't follow the argument. There must be additional processes at play to explain their data. One option is that continuous scavenging pulls the Ce/Ce* to HREE/LREE slope downward away from the apparent HTHH lava/pumice endmembers. Another is that volcanic ash has a distinct Ce/Ce* to HREE/LREE ratio from the lava and pumice.

Line by line comments:

Lines 80-82: Please add a citation for the scavenging signature on LREE/HREE patterns

Line 110: Add "A" before "Large" and change amount to amounts

Line 119 and Figure 1: I found this trend difficult to see on Figure 1, especially figure 1c. Could the stations be added to this plot with the air mass trajectories in Figure 1c? Even if they aren't labeled?

Line 145: At least a sentence discussing that scavenging and/or exchange processes must be occurring is needed here to maintain constant Nd concentrations while modifying the Nd isotopic composition – this is discussed in more detail later, but a sentence here would be helpful as well.

Line 151: This sentence doesn't make sense – scavenging must be playing an important role here in order to have the eNd signature persist as far east as it does even though Ce/Ce* drops precipitously in this same region (eastern portion of the shaded region in Figure 2).

Line 196-198: Seems like Australian dust range overlaps with the observed eNd

Line 295-297: Pavia et al. 2020 (cited later by the authors in this paper) derive an Fe residence time on the order of 0.75-6.5 years in the South Pacific Gyre – this would be inconsistent with the authors' argument for days to weeks residence times for Fe in this region, and I don't think their argument is correct

Lines 297-300: The authors do not have supporting evidence to say that scavenging is more important than biological uptake for Fe, or if they do, it isn't discussed to support this statement.

Section beginning on line 308: The authors include error bars on their Fe inputs from the HTHH eruption, but not Nd – the Nd error bars should be included. The authors should also include error bars on their estimate of the depositional area, which is not an error free determination.

There is a units problem lurking somewhere in this section, it seems. On line 328, the authors state there the HTHH eruption released 2.0×10^{15} g of ejecta, and calculate a potential release of 5.8×10^8 moles of Fe. First, the authors end up with a ~30% error, but there's multiple orders of magnitude error in the Fe fractional solubility estimate. How is this incorporated and/or why isn't it incorporated?

Lines 341-348: I think the authors have even more they could say about this – possibly connecting to Dunlea et al. 2015's estimates based on geochemical multivariate modeling that SPG sediments near the core top have 20-25% rhyolitic ash in them, possibly speculating on the role episodic Fe input (from volcanic eruptions) might play in setting variable nitrogen fixation in the SPG, which responds to short-term Fe supply (Guieu et al. 2014, Pavia et al. 2020) – this could cause a huge difference in long, time-integrated estimates of nitrogen fixation here (say, from inversions of nutrient distributions) compared to short-term nitrogen fixation rates derived from incubations.

Figure 3 legend: Change Astralian to Australian

Reviewer #2

(Remarks to the Author)

The manuscript submitted by Zhang et al. investigates the biogeochemical impact of the eruption of the Hunga Tonga-Hunga Ha'apai volcano in January 2022. In this study, the authors report surface concentrations and isotope composition of Neodymium (Nd), rare earth elements (REE) and trace metals (TMs) collected in the South Pacific Ocean to demonstrate the impact of the eruption on the surface ocean biogeochemistry. The dataset presented in this manuscript allows clearly demonstrating the link between the HTHH volcano eruption and change - over a large area - in Nd and REE characteristics. However, due to the time lag between the eruption and sample collection (~1-3 months), the trend in TMs concentrations is less clear. Furthermore, the link between TMs inputs and a biological response is not convincing, reducing the overall impact of this study.

That said, the authors have done a great job to demonstrate Nd and REE inputs from this volcanic event over a large area. Overall, it is an interesting study and I have only few concerns that need to be addressed (see below).

While a short-term biological response has been observed using satellite observations few days after the eruption (e.g. Barone et al. 2023), the authors claim that "Increasing trace metal concentrations in surface waters and chlorophyll-a inventories in euphotic layers between the eastern and western SPG further suggest that the volcanic eruption supplied (micro)nutrients stimulating a biological response" (L. 23-26).

I think that this statement of a long-term biological response (+1-3 months) is too speculative. It is mainly based on the relationship between the Chl-a inventory and surface Nd (Fig. 6f). However, the authors did not report any anomalies in Chl-a inventory relative to the pre-eruption period. Second, multiple sources of TMs are known in the study area, especially shallow hydrothermal vents that supply large amount of dissolved iron to the area, sustaining large phytoplankton blooms at the regional scale (Bonnet et al. 2023).

A map showing the dispersion of the ash particles in the surface waters 1 to 3 months after deposition (the period studied here) would be much more useful than the map presented in Fig. 1b showing ash deposition 72-h after the eruption. A relationship between the volcanic ash distribution (after 1-3 month of dispersion) vs. Nd, Ce and/or REE anomalies would be helpful to show that the eruption and ash deposition represent the main mechanism responsible the Nd and REE anomalies observed in the area after 1-3 months.

Table 1 – Nd, Ce anomaly and HREE enrichment values for shallow hydrothermal vents are missing in Table 1 (while stated L158-162). I am wondering if it's an oversight, or if no data has been published yet? Please specify.

Line 314-316. The estimation of the Nd input in the surface mixed layer requires vertical profiles of Nd in this 0-50 m layer to assess the vertical distribution of Nd.

Fig. 3. Please add the details of the linear model in the caption.

All the figures. Please define the depth range considered in the caption.

L205. "increasing" instead of "increase".

References

Barone, B., Letelier, R. M., Rubin, K. H. & Karl, D. M. Satellite Detection of a Massive Phytoplankton Bloom Following the 2022 Submarine Eruption of the Hunga Tonga-Hunga Ha'apai Volcano. *Geophys. Res. Lett.* 49, e2022GL099293 680 (2022).

Bonnet, S. et al. Natural iron fertilization by shallow hydrothermal sources fuels diazotroph blooms in the ocean. *Science*, 380, 812–817 (2023).

Reviewer #3

(Remarks to the Author)

Dear authors,

I have enjoyed reading a well written article presenting your results from the HTHH eruption.

The article has a great Nature Communications vibe, where the science meets a nice blend with communication to a more diverse public, thanks to its well written and thoughtful process.

Nature Comms provides us reviewers with some questions and concerns which we should think about. In your case, the methodology is widely accepted and has a strong backing.

I have made some short comments within the pdf file I was able to download from the application. I will also proceed to list them below.

L44: I do not like the term "ultra-oligotrophic".

I agree with the comment that the surface water of the SPG is the most oligotrophic in global oceans, with the lowest sea surface chl a concentrations (i.e. McClain et al., 2004; Morel et al., 2007). But the word oligotrophic already defines the lack of nutrients.

L52: I have found some references stating 5 and some stating 6. Please check.

L83: Change "Ce" to "cerium, Ce" like you did in L31 for Fe and Mn.

L102: Please check if you also need to add a reference to: SO289

L159: Could it possibly be from ongoing submarine eruptions and hydrothermal activity i.e. during a 2018 cruise in the same area, we observed submarine bubbling in numerous sites, with bubbles reaching the surface in many locations.

I guess not if we consider comment from L181? But with so many sources can we ever be certain?

L224: Please round it to 600-2400 km. If you want, here you could comment on the signal compared to other trace metal i.e. Fe in the EPR from Fitzsimmons or in the South Atlantic by Saito

L297: This was also determined in hydrothermal plumes i.e. González-Santana et al., (2020) and Rusakov (2007).

L306: Could it be a normal bloom in the area like the one observed by Tilliette et al., 2022 (i.e. Fig. 1)?

L315: As an English as a second language reader, I will need a bit of help with this unit. Would you please refer to it as pmol/kg of SW or as pmol/kg of pumice (or ash).

L326: The volcano will source Fe as Fe(II) which should be soluble (both in the aerial deposition and in the emission with the lava in contact with SW). This Fe is thought to be more bioavailable (at least it requires less energy to capture).

L331: I think this should be "potentially" or "could", since there could be sorption associated to organic compounds attaching to the pumice.

L382: Any reason of not filtering on-line directly from the teflon pump? So as to not have an extra step which could cause contamination (we use a T piece where one of the exits contains no filter but a valve where we can increase or decrease the flux through the second exit which contains the filter).

L452: Do you also have the information for Limits of Detection?

L455: Any idea on why this 1% continuous displacement?

L471: Old nomenclature, change to "ISO class 5", page 59 of the GEOTRACES cookbook.

L494: Why did you use water from another oceanic basin? There are potential changes caused by the water composition.

Cheers

David González-Santana

Version 1:

Reviewer comments:

Reviewer #1

(Remarks to the Author)

I am satisfied with the authors' revision and feel the manuscript is ready for publication.

Reviewer #2

(Remarks to the Author)

As stated in my first review report, this study brings important new insights into the impact of volcanic eruption on the surface ocean biogeochemistry. The authors have responded to all my comments and concerns. I therefore recommend publication of this study in Nature Communication.

Responses to reviewers for the manuscript: “Substantial trace metal input from the 2022 Hunga Tonga-Hunga Ha’apai eruption into the South Pacific Gyre” by Zhang et al. Please note that responses to reviewer comments are provided in blue below. All line numbers in responses refer to those in the tracked change version of the revised manuscript. References added during the revision process are highlighted in yellow in the revised reference list.

REVIEWER COMMENTS

Reviewer #1 (Remarks to the Author):

I enjoyed reading the manuscript by Zhang et al. on trace metal input from the 2022 Hunga Tonga-Hunga Ha’apai eruption to the South Pacific Gyre. The dataset is impressive, experimental design was strong, and the manuscript was well-written. I recommend moderate revisions but feel that the paper is publishable in Nature Communications – and indeed that the authors have some connections to draw at the end of the paper to make its impact even higher.

The difference in eNd and trace metals before and after the eruption was striking. I feel convinced that indeed, this eruption altered the REE and trace metal budgets of the surface of the SPG.

We thank the reviewer for their positive evaluation and constructive comments.

I had one most significant issue reading the paper. It’s discussed in lines 218-228 and Figure 3. It seems like there’s a problem with the authors’ proposed hypothesis of scavenging followed by volcanic material dissolution, because the trend of the data doesn’t lead towards either the HTHH lava or the pumice endmember. The authors did not really explain why this is the case, or if they did I didn’t follow the argument. There must be additional processes at play to explain their data. One option is that continuous scavenging pulls the Ce/Ce* to HREE/LREE slope downward away from the apparent HTHH lava/pumice endmembers. Another is that volcanic ash has a distinct Ce/Ce* to HREE/LREE ratio from the lava and pumice.

We agree with the reviewer that the trend of the data does not pass exactly through either the HTHH lava nor the pumice endmember, and that additional processes may be considered to explain the data. We regret that this was not addressed in the initially submitted version.

As the reviewer proposed, the preferential scavenging of LREE will result in an increase of the HREE/LREE ratio, which is reflected by a less steep slope of the Ce/Ce* versus HREE/LREE data (Fig.3). This is also consistent with our proposed hypothesis of Nd scavenging following volcanic material dissolution, given that Nd is one of the LREE.

We consider the second mechanism proposed by the reviewer, namely that volcanic ash has a Ce/Ce* to HREE/LREE ratio distinct from the lava and pumice, is less likely the correct explanation. Given that both the volcanic ash and pumice originate from the same source, a closely similar chemical composition can be expected.

In light of our observation of Nd scavenging following the dissolution of volcanic material, we favor the first mechanism put forth by the reviewer. This explanation has now also been incorporated into the main text (Line 252-255):

“The preferential removal of Nd and other LREEs during particle scavenging, following the dissolution of volcanic material, resulted in a downward shift in the Ce/Ce to HREE/LREE slope.*

This explains the deviation of the linear regression between Ce anomaly and HREE enrichment from the pumice endmember (Fig. 3)."

Line by line comments:

Lines 80-82: Please add a citation for the scavenging signature on LREE/HREE patterns

A reference to the publication "Elderfield, H. and Greaves, M.J. "The rare earth elements in seawater." Nature 296.5854 (1982): 214-219." has been added (Line 83, 85).

Line 110: Add "A" before "Large" and change amount to amounts

Revised (Line 110).

Line 119 and Figure 1: I found this trend difficult to see on Figure 1, especially figure 1c. Could the stations be added to this plot with the airmass trajectories in Figure 1c? Even if they aren't labeled?

We used a purple line to illustrate the cruise track in Figure 1c, hence we consider it unnecessary to add the stations to this small sub-figure. The airmass trajectories overall demonstrate an eastward movement of the airmass providing evidence of the eastward dispersion of volcanic ash. This supports the argument that the volcanic ash may have influenced a large area of the SPG surface waters towards the east, which may also have been captured by our cruise.

Line 145: At least a sentence discussing that scavenging and/or exchange processes must be occurring is needed here to maintain constant Nd concentrations while modifying the Nd isotopic composition – this is discussed in more detail later, but a sentence here would be helpful as well.

We thank the reviewer for this suggestion. Two sentences have been added to this section (Line 149-154) to provide a brief explanation of this observation:

"However, surface water Nd concentrations in the western SPG remained consistently low (~4 pmol/kg), at levels comparable to previous measurements^{36,37} (Fig. 2b). It can be posited that scavenging and/or exchange processes occurred to maintain a constant Nd concentration while modifying the Nd isotopic composition. This will be discussed in greater detail in section "Rapid release and scavenging of Nd following the volcanic input"."

Line 151: This sentence doesn't make sense – scavenging must be playing an important role here in order to have the eNd signature persist as far east as it does even though Ce/Ce* drops precipitously in this same region (eastern portion of the shaded region in Figure 2).

We agree with the reviewer here that scavenging also played a role following the dissolution of volcanic material, as also outlined in our response to the reviewer's major concern above. The sentence in question (Line 159-162) has now been deleted, and a discussion of the scavenging processes has been incorporated into the discussion (lines 252-255).

Line 196-198: Seems like Australian dust range overlaps with the observed eNd

The ϵNd range of Australian dust (-29 to +2) exhibits a very small overlap with the observed surface water ϵNd signatures (-0 to +1), due to the presence of samples from the Darling sub-basin (ϵNd ~-5 to +2), which, when considered in aggregate, still displays a negative value in comparison to the

positive surface seawater signal found in the western SPO. The ϵNd value for dust from all other areas of Australia is significantly lower than -2 due to the prevalence of Precambrian source rocks.

Even with the upper bound of the ϵNd value ($+2$), assuming an initial surface seawater Nd concentration of ~ 4 pmol/kg with an ϵNd value ~ -2 in the western SPG, an addition of 12 pmol/kg of Nd from the dust with an ϵNd value of $+2$ is required to obtain a surface ϵNd value of $+1$. This is, however, considered highly unlikely given that the observed Nd concentrations remained at a relatively low level.

Furthermore, the REE data do not corroborate a significant source contribution from the dust (Fig. 3) and, as discussed in Lines 208-211, the western part of the transect is located at a distance from the region receiving long-range dust transport from Australia, which follows a southeastward trajectory (Wengler et al., 2019).

Line 295-297: Pavia et al. 2020 (cited later by the authors in this paper) derive an Fe residence time on the order of 0.75-6.5 years in the South Pacific Gyre – this would be inconsistent with the authors' argument for days to weeks residence times for Fe in this region, and I don't think their argument is correct

In the estimation of Pavia et al. 2020, the Fe residence time is considerably shorter at the edge of the gyre (station 2; 0.75-2.0 years) compared to the centre of the gyre (station 8; 2.0-6.5 year). This discrepancy can be attributed to higher dust-derived dFe flux observed at the edge of the gyre. This indicates that in areas with high ash deposition, a large depositional flux of particles will result in a shorter dFe residence time.

The short residence times, with timescales of days to weeks, mentioned in Lines 310-312, correspond to dFe residence times reported for Atlantic surface waters that receive enhanced atmospheric inputs from the Saharan desert (Croot et al., 2004; Sarthou et al., 2003). Our intention was to utilize these examples to posit the potential for a short residence time of dFe in the main ash deposition area.

Lines 297-300: The authors do not have supporting evidence to say that scavenging is more important than biological uptake for Fe, or if they do, it isn't discussed to support this statement.

We agree with the reviewer and have changed the sentence to (Lines 312-315):

“Consequently, the more pronounced depletion of dFe in the main ash-deposition area (Fig. 6b), despite the anticipated high input fluxes, is primarily a result of scavenging subsequent to the release and/or biological removal.”

Section beginning on line 308: The authors include error bars on their Fe inputs from the HTHH eruption, but not Nd – the Nd error bars should be included. The authors should also include error bars on their estimate of the depositional area, which is not an error free determination.

We concur with the reviewer and thank for this suggestion. An error bar has now been included for the Nd flux considering both the uncertainty in the concentration of released Nd and the uncertainty in the size of the depositional area.

- 1) In evaluating the uncertainty of the concentration of the release Nd, the variability of the measured dissolved Nd isotope composition in the western SPG needs to be taken into consideration. Assuming an initial surface seawater Nd concentration of ~ 4 pmol/kg with an

ϵNd value ~ -2 in the western SPG, an addition of 0.9-1.9 pmol/kg of Nd from the volcanogenic material with an ϵNd value of +7.5 is required to obtain a surface ϵNd signature of ~ 0 to +1. Lines 245-250 are revised accordingly.

- 2) The NOAA HYSPLIT volcanic ash dispersion model indicates a mass loading area coverage of 5.4×10^6 km² three days after the eruption (Fig. 1b), which can be considered to represent the lower boundary of the ash deposition area. At the same time, our data demonstrate a significant increase of 2-3 ϵNd units between stations 33 and 44 (152°W to 172°E) in the western SPG. Based on this longitudinal coverage and the latitudinal range of 30° (20°S to 50°S) indicated by the HYSPLIT model (Fig. 1b), we arrive at a maximum extent of the deposition area of 1.2×10^7 km². Lines 334-344 are revised accordingly.

Accordingly, the range of the estimated Nd concentration increase as a consequence of the release (0.9-1.9 pmol/kg) combined with that of the ash deposition area (0.54 - 1.2×10^7 km²) indicates that the ash-derived Nd flux was likely between 0.4 - 1.6×10^8 g. Lines 344-346 are revised accordingly.

There is a units problem lurking somewhere in this section, it seems. On line 328, the authors state there the HTHH eruption released 2.0×10^{15} g of ejecta, and calculate a potential release of 5.8×10^8 moles of Fe. First, the authors end up with a $\sim 30\%$ error, but there's multiple orders of magnitude error in the Fe fractional solubility estimate. How is this incorporated and/or why isn't it incorporated?

The potential release of Fe is estimated on the basis of a comprehensive assessment of the global mean Fe release of 200 ± 50 nmol Fe per gram of ash for subduction zone volcanic ash, as reported by Olgun et al. (2011). This estimation represents the amount of Fe that can be mobilized/solubilized upon contact with seawater per gram of ash. It is suggested that this figure is largely independent of the bulk composition of the ash. The estimated fractional Fe solubility of 0.003-0.2% was derived from their estimation of the Fe release and is not relevant to our calculation.

We have now omitted citing the fractional solubility of iron to avoid any potential confusion (Line 356).

Lines 341-348: I think the authors have even more they could say about this – possibly connecting to Dunlea et al. 2015's estimates based on geochemical multivariate modeling that SPG sediments near the core top have 20-25% rhyolitic ash in them, possibly speculating on the role episodic Fe input (from volcanic eruptions) might play in setting variable nitrogen fixation in the SPG, which responds to short-term Fe supply (Guieu et al. 2014, Pavia et al. 2020) – this could cause a huge difference in long, time-integrated estimates of nitrogen fixation here (say, from inversions of nutrient distributions) compared to short-term nitrogen fixation rates derived from incubations.

We thank the reviewer for the suggestion to expand the implications of our budget calculation of trace metals.

Establishing a link between large-scale iron budgets and nitrogen fixation rates has proven challenging so far. For example, the amount of nitrogen fixation facilitated by HTHH Fe input can be estimated using the same calculation as the one employed by Pavia et al. (2020), in which the authors estimated the nitrogen fixation rate stimulated by dust input to the SPG:

We assume all the solubilized/released Fe ($5.8 \pm 1.5 \times 10^8$ mol) from the HTHH eruption is bioavailable. As illustrated in Pavia et al. (2020), using cellular stoichiometries of Crocosphera as a representative of the dominant diazotroph in the SPG for (Fe in nitrogenase):C and C:N, we can derive an upper bound on N_2 -fixation (N_{fix}) that can be supported by a given Fe flux:

$$N_{fix} = F_{Fe,bio} * \frac{\text{moles cellular C}}{\mu\text{mol Fe in Nitrogenase}} * \frac{\text{moles cellular N}}{\text{moles cellular C}}$$

$$= 5.8 \pm 1.5 \times 10^8 \text{ mol} * \frac{1 \text{ mol}}{15.8 \mu\text{mol}} * \frac{1 \text{ mol}}{8.6 \text{ mol}} = 4.1 \pm 1.1 \times 10^{11} \text{ mol} = 60 \pm 15 \text{ Tg N}$$

However, a number of the parameters employed in this calculation are subject to significant uncertainty, including the fraction of bioavailable erupted Fe, the dominant N₂-fixing organisms present, the proportion of Fe input that is bound in nitrogenase, the cellular stoichiometries, etc.

We therefore decided to not extend our estimation of Fe flux from the HTHH eruption to the nitrogen fixation rate, given the considerable uncertainty involved and the speculative nature of any conclusions that might be drawn without further data on nitrogen fixation itself. Delving deeply into the implications for nitrogen fixation is beyond the scope of the current manuscript. Further work is required to gain a detailed understanding of the response of nitrogen fixation to such episodic volcanic input, in order to estimate the large-scale integrated nitrogen fixation rate. Nevertheless, we have added a sentence at the end of this section:

“In order to improve the accuracy of biogeochemical models in the Pacific Ocean, it is therefore necessary to incorporate the episodic volcanic input of trace metals and to improve parameterization of the key chemical-biological processes that respond to volcanic inputs in the models.” (Line 379-382)

Figure 3 legend: Change Astralian to Australian

Sorry for the mistake. We have fixed it.

Reviewer #2 (Remarks to the Author):

The manuscript submitted by Zhang et al. investigates the biogeochemical impact of the eruption of the Hunga Tonga-Hunga Ha’apai volcano in January 2022. In this study, the authors report surface concentrations and isotope composition of Neodymium (Nd), rare earth elements (REE) and trace metals (TMs) collected in the South Pacific Ocean to demonstrate the impact of the eruption on the surface ocean biogeochemistry. The dataset presented in this manuscript allows clearly demonstrating the link between the HTHH volcano eruption and change - over a large area - in Nd and REE characteristics. However, due to the time lag between the eruption and sample collection (~1-3 months), the trend in TMs concentrations is less clear. Furthermore, the link between TMs inputs and a biological response is not convincing, reducing the overall impact of this study.

That said, the authors have done a great job to demonstrate Nd and REE inputs from this volcanic event over a large area. Overall, it is an interesting study and I have only few concerns that need to be addressed (see below).

We thank the reviewer for the positive comment on our work.

It is acknowledged that the time lag between the eruption and sample collection in the western SPG (9-10 weeks) likely resulted in a less clear trend in TMs concentrations, due to scavenging and/or biological uptake. However, as shown in Figure S7, a comparison of surface trace metal concentrations in the western SPG between our western transect and the two earlier GEOTRACES cruises, GP13 and GP19, revealed elevated surface TMs concentrations in the region 175°E-175°W for our cruise, with a notable increase in dAl over an extended area. This documents that elevated TMs concentrations from the eruption were still present in the area during our cruise.

With regard to the biological response, we address this concern in the following two paragraphs.

While a short-term biological response has been observed using satellite observations few days after the eruption (e.g. Barone et al. 2023), the authors claim that “Increasing trace metal concentrations in surface waters and chlorophyll-a inventories in euphotic layers between the eastern and western SPG further suggest that the volcanic eruption supplied (micro)nutrients stimulating a biological response” (L. 23-26).

I think that this statement of a long-term biological response (+1-3 months) is too speculative. It is mainly based on the relationship between the Chl-a inventory and surface Nd (Fig. 6f). However, the authors did not report any anomalies in Chl-a inventory relative to the pre-eruption period. Second, multiple sources of TMs are known in the study area, especially shallow hydrothermal vents that supply large amount of dissolved iron to the area, sustaining large phytoplankton blooms at the regional scale (Bonnet et al. 2023).

We acknowledge that linking the chlorophyll-a concentrations measured on our research cruise to eruption-derived micronutrient supply is challenging and subject to caveats and we felt that this was to some extent reflected in the wording in our initial manuscript. In order to add some additional support to the statements made, we conducted a further analysis of satellite-derived chlorophyll-a for the cruise period (March 2022) relative to a multi-decade March climatological average. A map of chlorophyll-a anomalies is provided below, which shows the relative change in chlorophyll-a concentrations in March 2022 compared to the March average for the period 2002 to 2023 (i.e., $(\text{March}_{2022} - \text{March}_{\text{Average}}) / \text{March}_{\text{Average}}$). As illustrated in the map, a broad region with a positive chlorophyll-a anomaly is observed along the western transect overlapping with our western stations, where the most significant volcanic input signal was identified based on geochemical evidence. This map has now been included in the supplementary material as additional support for our statement that micronutrient supply from the eruption may have led to enhanced chlorophyll-a concentrations (Fig. S8, Line 318-320):

“In addition, a positive satellite-derived Chl-a anomaly for March 2022 was evident in the western part of the transect compared to the March average for the period between 2002 and 2023 (Fig. S8).”

Figure S8. Map of satellite-derived chlorophyll-a anomalies showing the relative change in chlorophyll-a concentrations in March 2022 compared to the March average for the period 2002 to 2023 in the South

Pacific Ocean. The satellite product employed was Aqua-MODIS Chlorophyll-a from NASA (<https://oceancolor.gsfc.nasa.gov/>).

Our statement in the main text reads “*Together these indicate a **potential** increase in phytoplankton growth in response to the volcanic input of dFe...*” (Lines 320-321), which we feel reflects uncertainty in interpreting the observed chlorophyll-a variability. Furthermore, we have added a discussion to further emphasize the uncertainty in the main text (Lines 324-330):

“Given that multiple sources of trace metals are known in the western SPG, confidently ascribing this elevated phytoplankton biomass to volcanic dFe fertilization over and above other sources is challenging. However, as discussed above, volcanic input was identified as the dominant TM source to surface waters, therefore we consider a linkage between volcanic input from the HTHH eruption into the western SPG and the enhanced TChl-a inventory a plausible scenario.”

In addition, and for the reasons outlined above, the sentence in the abstract related to the biological response has been amended to include the term "potentially" (Line 26):

*“Increasing trace metal concentrations in surface waters and chlorophyll-a inventories in euphotic layers between the eastern and western SPG further suggest that the volcanic eruption supplied (micro)nutrients **potentially** stimulating a biological response.”*

A map showing the dispersion of the ash particles in the surface waters 1 to 3 months after deposition (the period studied here) would be much more useful than the map presented in Fig. 1b showing ash deposition 72-h after the eruption. A relationship between the volcanic ash distribution (after 1-3 month of dispersion) vs. Nd, Ce and/or REE anomalies would be helpful to show that the eruption and ash deposition represent the main mechanism responsible for the Nd and REE anomalies observed in the area after 1-3 months.

In addition to modelling the ash deposition 72 hours after the eruption, we have also modelled the air mass forward trajectories within two weeks after the HTHH eruption. This should represent the general direction of ash dispersion following the air masses and covers the maximum time period that the NOAA HYSPLIT model is capable of addressing. It is not possible to run the model for an extended period of time, namely between one and three months.

A map estimating the dispersion of the ash particles in surface ocean waters 1 to 3 months after deposition would require the use of a more complex coupled atmospheric-ocean model, which is beyond the scope of this work. In light of our extensive discussion on potential external inputs to SPG surface waters, it can be concluded with a high degree of confidence that the volcanic input from the HTHH eruption represents the main mechanism responsible for the radiogenic Nd isotope signatures and heavy REE enrichment in the western SPG.

Table 1 – Nd, Ce anomaly and HREE enrichment values for shallow hydrothermal vents are missing in Table 1 (while stated L158-162). I am wondering if it's an oversight, or if no data has been published yet? Please specify.

The chemical compositions of fluids from different submarine hydrothermal systems along the Kermadec arc show a large variability, both spatially and temporally (Kleint et al., 2019). Nd isotope composition and REE patterns of vent fluids from shallow hydrothermal vent fluids at the Kermadec arc are not available, which is why this item is not included in Table 1.

Line 314-316. The estimation of the Nd input in the surface mixed layer requires vertical profiles of Nd in this 0-50 m layer to assess the vertical distribution of Nd.

The vertical distribution of Nd isotopes and concentrations within the mixed layer is generally uniform due to rapid mixing which homogenizes physical and chemical properties. In other words, due to wind-driven mixing, the mixed layer generally shows uniform concentrations of nutrients and trace metals. Consequently, high-resolution large volume water sampling for Nd isotopes and corresponding Nd concentrations throughout the mixed layer was deemed unnecessary and only surface water samples were taken from the mixed layer.

Fig. 3. Please add the details of the linear model in the caption.

This information has been added to the caption of Fig.3 (Lines 621-622):

*“The dashed blue line and dotted blue lines represent the linear regression of Ce/Ce^*_{PAAS} and $HREE/LREE_{PAAS}$ in the surface water of the South Pacific Gyre (this study) and the 95% confidence interval of the regression, respectively.”*

All the figures. Please define the depth range considered in the caption.

The information has been added.

L205. “increasing” instead of “increase”.

We think that it should say “increase” here, as it is a verb (Line 214).

References

Barone, B., Letelier, R. M., Rubin, K. H. & Karl, D. M. Satellite Detection of a Massive Phytoplankton Bloom Following the 2022 Submarine Eruption of the Hunga Tonga-Hunga Ha’apai Volcano. *Geophys. Res. Lett.* 49, e2022GL099293 680 (2022).

Bonnet, S. et al. Natural iron fertilization by shallow hydrothermal sources fuels diazotroph blooms in the ocean. *Science*, 380, 812–817 (2023).

Reviewer #3 (Remarks to the Author):

Dear authors,

I have enjoyed reading a well written article presenting your results from the HTHH eruption. The article has a great Nature Communications vibe, where the science meets a nice blend with communication to a more diverse public, thanks to its well written and thoughtful process. Nature Comms provides us reviewers with some questions and concerns which we should think about. In your case, the methodology is widely accepted and has a strong backing. I have made some short comments within the pdf file I was able to download from the application. I will also proceed to list them below.

We are pleased with the positive assessment of this article by the reviewer.

L44: I do not like the term "ultra-oligotrophic".

I agree with the comment that the surface water of the SPG is the most oligotrophic in global oceans, with the lowest sea surface chl a concentrations (i.e. McClain et al., 2004; Morel et al., 2007). But the word oligotrophic already defines the lack of nutrients.

The term "ultra-oligotrophic" has been modified to "oligotrophic" (Line 45).

L52: I have found some references stating 5 and some stating 6. Please check.

We only find one peer-reviewed article (Terry et al., 2022) indicating a VEI of 5, which was referred to a YouTube video "Jan 2022 eruption of Hunga Volcano, Tonga - first insights. by Shane Cronin".

In the Global Volcanism Program (Smithsonian Institution), the VEI of the 2022 January 15 eruption is reported as 5 (<https://volcano.si.edu/volcano.cfm?vn=243040&vtab=Eruptions>). However, it is noted that this estimation is "*based on news reports/interviews describing work by Cronin et al.; will revise if needed based on final published maps or papers.*"

Accordingly, we are inclined to cite the VEI estimate (5.8) from the peer-reviewed article (Poli and Shapiro, 2022), which is based on analyses of seismic data from global seismic networks.

L83: Change "Ce" to "cerium, Ce" like you did in L31 for Fe and Mn.

Revised (Line 83)

L102: Please check if you also need to add a reference to: SO289

We thank the reviewer for this reminder! It has been added now (Line 103).

L159: Could it possible be from ongoing submarine eruptions and hydrothermal activity i.e. during a 2018 cruise in the same area, we observed submarine bubbling in numerous sites, with bubbles reaching the surface in many locations.

I guess not if we consider comment from L181? But with so many sources can we ever be certain?

Submarine bubble plumes, often occur above active submarine hydrothermal systems and cold seeps, are formed by bubbles of free gas rising through the water column. These are distinct from the floating tephra plume (e.g., pumice) that we observed in the surface water. Pumice is commonly produced by the eruption of explosive volcanoes, and is created when super-heated, highly pressurized rock is rapidly ejected from a volcano.

Indeed, silicic effusive eruptions have also been observed to produce pumice rafts, as for example was the case during the 2012 Havre submarine eruption. However, as has been clearly discussed and depicted in Manga et al. (2018) about the effusive submarine eruptions:

"Large clasts (>1 m) rose to the sea surface without ingesting enough water to sink. Those large clasts with sufficient isolated vesicles and/or trapped gas remained afloat in the raft. Smaller clasts would

not have reached the surface, ingesting water quickly and settling close to the vent, or were transported by currents if small enough” (illustrated in figure below)

The pumice we observed, collected, and measured were small clasts (sizes of up to several centimeters). Thus, it is more probable that they originated from a subaerial tephra plume and were produced by an explosive volcanic eruption.

[REDACTED]

L224: Please round it to 600-2400 km. If you want, here you could comment on the signal compared to other trace metal i.e. Fe in the EPR from Fitzsimmons or in the South Atlantic by Saito

Revised as suggested (Line 233).

We thank the reviewer for suggesting such comparison. Nevertheless, we consider it unnecessary to make a comparison between the redistribution of volcanic input by surface currents and the redistribution of mid-ocean ridge hydrothermal input by ocean circulation as these are two distinct processes.

L297: This was also determined in hydrothermal plumes i.e. González-Santana et al., (2020) and Rusakov (2007).

We thank the reviewer for this suggestion. However, the intention here is to create an analogue for the Fe residence time in the surface mixed layer subsequent to intensive ash deposition (Lines 310-312). Therefore, we consider it inappropriate to use examples from hydrothermal plumes as analogues in this context.

L306: Could it be a normal bloom in the area like the one observed by Tilliette et al., 2022 (i.e. Fig. 1)?

As discussed in Lines 213-226, no geochemical evidence for significant shallow hydrothermal inputs near the Tonga-Kermadec arc was identified during our cruise. Therefore, we consider a linkage between volcanic input from the HTHH eruption into the western SPG and the enhanced TChl-a inventory more likely. However, we cannot attribute the increase in Chl-a inventory exclusively to volcanic input, nor ascribe this elevated phytoplankton biomass to volcanic dFe fertilization over and above other sources. Accordingly, we have toned down our assertion on the linkage between volcanic input and the biological response. For further details, please refer to our response to Reviewer 2's major comment.

L315: As an English as a second language reader, I will need a bit of help with this unit. Would you please refer to it as pmol/kg of SW or as pmol/kg of pumice (or ash).

Here it means “of seawater”. This information has now been added (Line 247).

L326: The volcano will source Fe as Fe(II) which should be soluble (both in the aerial deposition and in the emission with the lava in contact with SW). This Fe is thought to be more bioavailable (at least it requires less energy to capture).

The fractional Fe solubility of 0.003-0.2% was estimated by Olgun et al. (2011) based on geochemical experiments conducted to determine the rapid release of Fe upon contact of pristine volcanic ash with seawater. In response to a comment from Reviewer 1, we have deleted this information (Line 356).

L331: I think this should be "potentially" or "could", since there could be sorption associated to organic compounds attaching to the pumice.

We have changed “likely” to “potentially” (Line 362).

L382: Any reason of not filtering on-line directly from the Teflon pump? So as to not have an extra step which could cause contamination (we use a T piece where one of the exits contains no filter but a valve where we can increase or decrease the flux through the second exit which contains the filter).

The online filtration of trace metals (Fe, Mn, Al, etc.) is described in a later section. This section focuses on the sampling of large-volume samples for Nd isotope measurements (20-40L), which necessitate a lengthy online filtration process. Additionally, Nd is less susceptible to contamination than other trace metals, as evidenced by the low whole-procedure blank for Nd concentration (Line 439). The offline filtration of Nd isotope samples is a widely accepted procedure, as outlined in the GEOTRACES protocol (Cutter et al., 2010).

L452: Do you also have the information for Limits of Detection?

Yes. We have added the Limits of Detection for all REEs to the dataset (Supplementary Data 1).

L455: Any idea on why this 1% continues displacement?

Isotope dilution combined with MC-ICP-MS and Sea-Fast preconcentration coupled with ICP-MS are two distinct methods to measure Nd concentrations. The former method is more precise and exhibits a significantly smaller error (2SD<0.4%) compared to the latter (2SD=7.6%). However, the latter method is capable of measuring the concentrations of all rare earth elements simultaneously.

The objective of the comparison between these two different methods is to demonstrate the accuracy of our measurements and to evaluate the precision of the REE measurements based on the Sea-Fast preconcentration method. Our results show that the REE measurements are precise, with a difference of only 1% between the two methods. The discrepancy of 1% is estimated based on a linear regression model and is attributed to the inherent differences between the two methods. This small discrepancy may be caused by a 1% evaporation during the SeaFast method or may be attributable to other factors.

L471: Old nomenclature, change to "ISO class 5", page 59 of the GEOTRACES cookbook.

Revised (Line 506).

L494: Why did you use water from another oceanic basin? There are potential changes caused by the water composition.

A large volume of the surface water from the South Atlantic Gyre has been stored in our home lab. It is typically what is used as the base seawater for the standard addition method of Al concentration measurement due to its low concentrations of trace metals. Using this seawater ensures the consistency of the matrix between samples and the standard curve.

Cheers

David González-Santana

References:

- Croot, P.L., Streu, P., Baker, A.R., 2004. Short residence time for iron in surface seawater impacted by atmospheric dry deposition from Saharan dust events. *Geophys. Res. Lett.* 31, L23S08. <https://doi.org/10.1029/2004GL020153>
- Cutter, G.A., Andersson, P., Codispoti, L., Croot, P., Place, P., Hoe, T., Kingdom, U., Francois, R., Sciences, O., Lohan, M., Circus, D., Obata, H., 2010. Sampling and Sample-handling Protocols for GEOTRACES Cruises.
- Elderfield, H., Greaves, M.J., 1982. The rare earth elements in seawater. *Nature* 296, 214–219. <https://doi.org/10.1038/296214a0>
- Kleint, C., Bach, W., Diehl, A., Fröhberg, N., Garbe-Schönberg, D., Hartmann, J.F., de Ronde, C.E.J., Sander, S.G., Strauss, H., Stucker, V.K., Thal, J., Zitoun, R., Koschinsky, A., 2019. Geochemical characterization of highly diverse hydrothermal fluids from volcanic vent systems of the Kermadec intraoceanic arc. *Chem. Geol.* 528, 119289. <https://doi.org/10.1016/j.chemgeo.2019.119289>
- Manga, M., Fauria, K.E., Lin, C., Mitchell, S.J., Jones, M.P., Conway, C.E., Degruyter, W., Hosseini, B., Carey, R., Cahalan, R., Houghton, B.F., White, J.D.L., Jutzeler, M., Soule, S.A., Tani, K., 2018. The pumice raft-forming 2012 Havre submarine eruption was effusive. *Earth Planet. Sci. Lett.* 489, 49–58. <https://doi.org/10.1016/j.epsl.2018.02.025>
- Olgun, N., Duggen, S., Croot, P.L., Delmelle, P., Dietze, H., Schacht, U., Óskarsson, N., Siebe, C., Auer, A., Garbe-Schönberg, D., 2011. Surface ocean iron fertilization: The role of airborne volcanic ash from subduction zone and hot spot volcanoes and related iron fluxes into the Pacific Ocean. *Global Biogeochem. Cycles* 25, GB4001. <https://doi.org/10.1029/2009GB003761>
- Poli, P., Shapiro, N.M., 2022. Rapid Characterization of Large Volcanic Eruptions: Measuring the Impulse of the Hunga Tonga Ha’apai Explosion From Teleseismic Waves. *Geophys. Res. Lett.* 49, e2022GL098123. <https://doi.org/10.1029/2022GL098123>
- Sarthou, G., Baker, A.R., Blain, S., Achterberg, E.P., Boye, M., Bowie, A.R., Croot, P., Laan, P., de Baar, H.J.W., Jickells, T.D., Worsfold, P.J., 2003. Atmospheric iron deposition and sea-surface dissolved iron concentrations in the eastern Atlantic Ocean. *Deep Sea Res. Part I Oceanogr. Res. Pap.* 50, 1339–1352. [https://doi.org/10.1016/S0967-0637\(03\)00126-2](https://doi.org/10.1016/S0967-0637(03)00126-2)
- Terry, J.P., Goff, J., Winspear, N., Bongolan, V.P., Fisher, S., 2022. Tonga volcanic eruption and tsunami, January 2022: globally the most significant opportunity to observe an explosive and tsunamigenic submarine eruption since AD 1883 Krakatau. *Geosci. Lett.* 9, 24. <https://doi.org/10.1186/s40562-022-00232-z>
- Wengler, M., Lamy, F., Struve, T., Borunda, A., Böning, P., Geibert, W., Kuhn, G., Pahnke, K., Roberts, J., Tiedemann, R., Winckler, G., 2019. A geochemical approach to reconstruct modern dust fluxes and sources to the South Pacific. *Geochim. Cosmochim. Acta* 264, 205–223. <https://doi.org/10.1016/j.gca.2019.08.024>